# RNA splicing is a key mediator of tumour cell plasticity and a therapeutic vulnerability in colorectal cancer

Adam E. Hall[1,2,7], Sebastian Öther-Gee Pohl [1,2,7], Patrizia Cammareri[1,2], Stuart Aitken [1,3], Nicholas T. Younger[4], Michela Raponi[1,2,5], Caroline V. Billard[1,2], Alfonso Bolado Carrancio[1,2] Aslihan Bastem[1,2], Paz Freile[1,2], Fiona Haward[1,3,6], Ian R. Adams [1,3], Javier F. Caceres [1,3] Paula Preyzner[1,2], Alex von Kriegsheim[1,2], Malcolm G. Dunlop[1,3], Farhat V. Din[1,2] & Kevin B. Myant [1,2✉]

Tumour cell plasticity is a major barrier to the efficacy of targeted cancer therapies but the mechanisms that mediate it are poorly understood. Here, we identify dysregulated RNA splicing as a key driver of tumour cell dedifferentiation in colorectal cancer (CRC). We find that *Apc*-deficient CRC cells have dysregulated RNA splicing machinery and exhibit global rewiring of RNA splicing. We show that the splicing factor SRSF1 controls the plasticity of tumour cells by controlling *Kras* splicing and is required for CRC invasion in a mouse model of carcinogenesis. *SRSF1* expression maintains stemness in human CRC organoids and correlates with cancer stem cell marker expression in human tumours. Crucially, partial genetic downregulation of *Srsf1* does not detrimentally affect normal tissue homeostasis, demonstrating that tumour cell plasticity can be differentially targeted. Thus, our findings link dysregulation of the RNA splicing machinery and control of tumour cell plasticity.

[1] Institute of Genetics and Cancer, The University of Edinburgh, Western General Hospital Campus, Crewe Road, Edinburgh EH4 2XU, Scotland. [2] Cancer Research UK Edinburgh Centre, Institute of Genetics and Cancer, The University of Edinburgh, Western General Hospital, Crewe Road South, Edinburgh EH4 2XR, Scotland. [3] MRC Human Genetics Unit, Institute of Genetics and Cancer, The University of Edinburgh, Western General Hospital, Crewe Road, Edinburgh EH4 2XU, Scotland. [4] Centre for Inflammation Research, University of Edinburgh, Edinburgh EH16 4TJ, Scotland. [5] Present address: Cancer Research UK Beatson Institute, Garscube Estate, Switchback Road, Bearsden, Glasgow G61 1BD, Scotland. [6] Present address: Centre for Gene Regulation & Expression, School of Life Sciences, University of Dundee, Dow Street, Dundee DD1 5EH, Scotland. [7] These authors contributed equally: Adam E. Hall, Sebastian Öther-Gee Pohl. ✉email: kevin.myant@igmm.ed.ac.uk

Colorectal cancer (CRC) is a leading cause of cancer related mortality responsible for around 900,000 deaths annually[1]. The emergence of novel targeted therapies for treatment has offered hope of improved outcome[2], but these are generally beneficial to only small subsets of patients and emergence of resistance is common, even in initially responsive patients. For example, resistance to the highly specific EGFR inhibitor cetuximab, used for the treatment of some KRAS wild-type tumours, routinely emerges and survival benefit is limited[3]. In this case resistance emerges via multiple, distinct genetic and transcriptomic mechanisms that bypass the tumour's requirement for signalling via EGFR, suggesting tumour heterogeneity may play a role in mediating therapy resistance[3]. Indeed, intra- and inter-tumoural mutational diversity within patients has been shown to result in CRC subpopulations having variable responses to therapeutic agents[4], together suggesting that targeting single specific oncogenic driver genes may not deliver durable responses, even in highly stratified patient populations. Therefore, there is an urgent need to identify more general, clinically targetable biological mechanisms that exploit general cancer cell characteristics rather than the classically studied oncogenic driver genes that mediate them, such mutated APC, TP53 or RAS/MAPK[5].

Another key therapeutic challenge is overcoming the plasticity of stem cell fate, where stem cells can not only differentiate into more specialised cell types, but these differentiated cells have the ability to revert back to the stem cell phenotype following tissue damage[6–8]. This is especially important when considering clinical treatment as this phenomenon has also been observed in tumours where differentiated tumour cells (non-Lgr5 + ) can be reactivated following cancer stem cell depletion and subsequently dedifferentiate and fuel tumour growth[9,10].

Alternative RNA splicing increases proteome diversity and can induce radical effects on cellular phenotype including promoting carcinogenesis[11–13]. For example, the alternatively spliced Rac1 isoform, Rac1b, is constitutively active and is upregulated in Wnt-driven CRC driving efficient tumourigenesis[14–16]. Additionally, mRNA splicing is thought to be the rate-limiting step in generating functional transcripts and has been proposed as a potential therapeutic vulnerability in MYC driven breast cancer and lymphoma[17–19]. Therefore, activity levels of the RNA spliceosome may be an important facilitator of cancer cell growth and its targeting may present a viable therapeutic strategy.

Here, we aimed to identify universal therapeutic targets for CRC by determining the mechanisms important for cancer cell growth. We identified widespread dysregulation of RNA splicing factors and global reprogramming of RNA splice isoforms in a Wnt-driven animal model of CRC. Using 3D organoid cultures, we performed a synthetic lethal screen of splicing factors altered in CRC, identifying the splicing factor SRSF1 as a therapeutic vulnerability in CRC organoids. Moreover, we determined that SRSF1, by modulating the pro-proliferative Kras 4B isoform, alters cancer stem cell plasticity, the invasive potential of advanced tumours and the viability of primary CRC organoids derived from patients. Together, these data suggest that RNA splicing, and SRSF1 in particular, control the plasticity of tumour cells and is a viable therapeutic target in CRC.

## Results

### RNA splicing is dysregulated following Wnt hyperactivation.
To identify cellular processes activated at early stages of intestinal carcinogenesis, we deleted both copies of the CRC tumour suppressor gene Apc in mice using the intestinal specific villin-Cre$^{ERT2}$ (villinCre$^{ERT2}$ Apc$^{fl/fl}$). Five days after tamoxifen induced-gene recombination, small intestinal tissue was dissected and activation of the Wnt hyperproliferative phenotype

confirmed by immunohistochemical staining of BrdU and Wnt/β-catenin target gene upregulation (Fig. S1a, S1b and S3a).

Wild-type and Apc$^{fl/fl}$ intestinal tissue was subjected to RNAseq (Supplementary Data 1) and biological pathway analysis (KEGG) used on differentially expressed transcripts to identify enriched biological processes. Compared to normal tissue, RNA processing events were the most significantly upregulated processes following Apc loss (Fig. 1a). Specifically, we found that 60 genes involved in RNA splicing were upregulated (Supplementary Data 2 and Fig. S1c). This suggests there may be an increased requirement for RNA splicing following CRC initiation.

To ascertain if splicing factor upregulation correlated with changes in RNA splicing events, we conducted differential splicing analysis using SUPPA2[20] and identified 1,661 alternative splicing events following Apc loss ($p < 0.05$) (Fig. 1b and Supplementary Data 3) indicating a global rewiring of alternative RNA splicing upon Wnt activation. The largest category of splicing event detected using this method was alternative first exon (46.4%) which could be an indication of alternative transcriptional regulation. To investigate this, we compared these events to differentially expressed genes identified in Supplementary Data 1. 140/631 alternative first exon events were also differentially expressed indicating the majority of these events are not linked to transcriptional changes and likely represent alternative use of first exons and/or differential promoter usage (Fig. S1d). SUPPA2 analysis detects annotated splicing events but is unable to detect unannotated events, limiting our analysis. To investigate whether loss of Apc also induces novel splicing events we carried out rMATS analysis, which can also detect novel splicing events[21]. Interestingly, this analysis identified a large number of splicing events utilising novel splice sites indicative of widespread splicing alterations (Supplementary Data 4 and Fig. S1e). Splicing changes were detected and validated in numerous cancer-related genes, including pro-tumourigenic Cd44 variant isoforms[22], Fgfr2 and eIF4A2 (Fig. 1c and S1f–S1h). Notably, we found a significant change in the splicing ratio of the oncogene Kras (Fig. 1c and S1f) favouring the Kras4B isoform over the Kras4A isoform. These isoforms have been previously reported to have pro-proliferative and proapoptotic effects respectively (Figure S1g)[23,24]. We next analysed previously published RNA splicing analysis of The Cancer Genome Atlas (TCGA) data[25] to determine whether the splicing events identified in our mouse model are found in human CRC samples. Of the top 100 identified skipped exon splicing events, 18 showed the same alteration in human CRC including KRAS, CD44 and EIF4A2 (Supplementary Data 5 and Fig. S1i). In summary, we identified widespread alterations of alternative splicing events following Apc loss, several of which are associated with cancer progression and a proportion of which are found in human CRC samples.

As RNA splicing activity is dysregulated in Apc$^{fl/fl}$ intestine, we investigated the effects of global splicing inhibition on oncogenic growth compared to wild-type. 3D intestinal organoids from normal or Apc$^{fl/fl}$ intestine were exposed to the highly potent splicing inhibitor pladienolide B[26] (Fig. 1d). At low concentrations of the drug, Apc$^{fl/fl}$ organoids showed signs of cell death (Fig. 1d) and a drop in cell viability (Fig. 1e and S1j), whereas wild-type organoids were more resilient. We found a significant difference in the IC$_{50}$ of pladienolide B for wild-type and Apc$^{fl/fl}$ intestinal organoids (Fig. 1e). We extended this analysis to organoids derived from human patient material. We found that CRC tumour organoids exhibited significant loss of viability upon pladienolide B treatment but organoids derived from normal tissue did not (Fig. 1f, g). These results support a proof-of-concept that the spliceosome may be a therapeutic vulnerability in Apc-deficient cells.

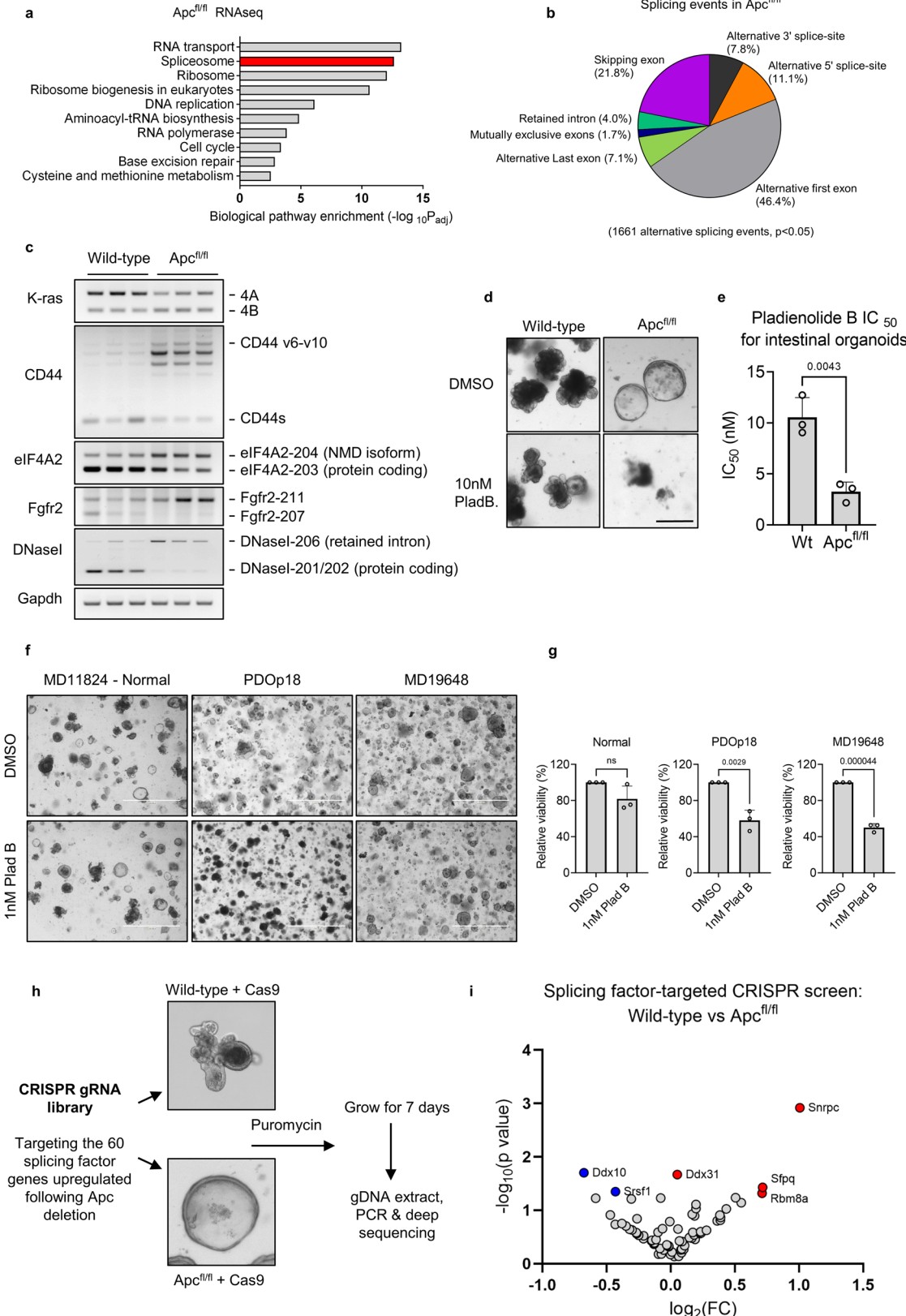

**The splicing factor SRSF1 is critical for *Apc*-deficient growth**. To identify which splicing factors are required for *Apc*-deficient cell growth, we conducted a splicing factor-targeted, synthetic lethal CRISPR screen comparing wild-type to *Apc*^fl/fl intestinal organoids. We generated a guide RNA (gRNA) library targeting the 60 splicing factor genes upregulated following *Apc* deletion

(Supplementary Data 2 and 6) and delivered them by lentiviral transduction into either wild-type-Cas9 or *Apc*^fl/fl-Cas9 intestinal organoids (Fig. 1h) at low multiplicity of infection (MOI). Surviving organoids were harvested after 7 days of antibiotic selection and gRNA sequences in surviving clones analysed via deep sequencing. A comparison of the gRNAs present in surviving

**Fig. 1 RNA splicing is dysregulated following Wnt hyperactivation. a** RNAseq performed on wild-type and $Apc^{fl/fl}$ mouse small intestines isolated 5 days post tamoxifen inductions and sequences were subjected to biological pathway enrichment analysis. **b** Alternative splicing events following $Apc$-deletion compared to wild-type. **c** RT-PCR validation of a selection of alternative splicing events. **d** Representative images of wild-type and $Apc^{fl/fl}$ intestinal organoids treated with the splicing inhibitor pladienolide B. Scale bar 250 μm. **e** Half maximal inhibitory concentration ($IC_{50}$) of pladienolide B in intestinal organoids, $n = 3$ independent experiments. **f** Representative images of organoids derived from normal human colon and 2 colonic tumours treated with 1 nM pladienolide B. Scale bar 1000 μm. **g** Quantification of viability assays following pladienolide B treatment, $n = 3$ independent experiments. **h** Schematic explanation of CRISPR screen design. **i** Guide RNA enrichment or depletion in surviving $Apc^{fl/fl}$ organoids compared to wild-type following completion of CRISPR screen. Data represented as mean and error bars SD. Data analysed with two-tailed, unpaired t-test, $p$ values are indicated in figure panels. See also Fig. S1.

wild-type-Cas9 and $Apc^{fl/fl}$-Cas9 organoids showed that guide RNAs targeting $Ddx10$ and $Srsf1$ were significantly under-represented in $Apc^{fl/fl}$-Cas9 population (Fig. 1i and Supplementary Data 7). This implies deletion of these genes is less well tolerated in the context of $Apc$ loss compared to normal growth conditions. To confirm these findings, we CRISPR-targeted $Ddx10$ and $Srsf1$ individually in pooled gene-edited populations of both wild-type and $Apc^{fl/fl}$ organoids. Targeting $Ddx10$ or $Srsf1$ reduced $Apc^{fl/fl}$ organoid growth, but only $Srsf1$ showed a significant reduction in viability in $Apc^{fl/fl}$ organoids compared to wild-type organoids where $Srsf1$ is deleted (Fig. 2a, b and S2a, S2b). Therefore, we chose to investigate the function of $Srsf1$ in further detail.

Notably, despite showing significantly reduced viability in $Apc^{fl/fl}$ organoids, deletion of $Srsf1$ by CRISPR also reduced viability in wild-type organoids. However, previous reports have demonstrated that modest reductions in $Srsf1$ expression levels are well tolerated in mice suggesting a potential therapeutic window for targeting $Srsf1^{27}$. As therapeutic targeting does not lead to target inhibition equivalent to genetic knockout we aimed to phenocopy clinical intervention more accurately using doxycycline-inducible short hairpin RNA (shRNA) expression. This also allowed us to deplete $Srsf1$ levels in fully developed organoids of increasing genetic complexity and aggressiveness. Expression of the $Srsf1$ shRNA reduced SRSF1 levels in wild-type, $Apc$-deficient and AKP ($Apc^{\Delta/\Delta}$ $Kras;^{G12D/+}$ $Trp53^{\Delta/\Delta}$) organoids (Figures S2c, S2d). Validating the findings with CRISPR deletion, knockdown of $Srsf1$ significantly reduced the growth of fully mature $Apc$-deficient and AKP organoids (Fig. 2e–h). However, we observed no impact of $Srsf1$ depletion on wild-type organoid growth (Fig. 2c, d). Together, these data suggest a potential therapeutic vulnerability of CRC cells to depletion of $Srsf1$.

**SRSF1 is required for epithelial cell hyperproliferation.** To investigate the effects of targeting $Srsf1$ in vivo, we combined $Apc$ deletion in our mouse model, where SRSF1 was upregulated (Fig. S3a), with a deletion of a single copy of $Srsf1$ ($villin$Cre$^{ERT2}$ $Apc;^{fl/fl}$ $Srsf1^{fl/+}$) (Figs. S3b–S3c and S3h). Homozygous deletion of $Srsf1$ led to gut toxicity with elevated levels of intestinal cell apoptosis (Fig. S3k) so heterozygous deletion of $Srsf1$ was selected to phenocopy a more clinically relevant scenario where impaired SRSF1 activity might be achieved rather than complete deletion.

A 30–50% reduction in $Srsf1$ levels (Figure S3h) had no proliferative or morphological effect in wild-type intestinal tissue (Fig. 3a, b and S3d–S3e). However, we observed a significant reduction in the number of BrdU-labeled (Fig. 3a, b) and Ki-67-positive cells (Figs. S3d, S3e) upon deletion of one copy of $Srsf1$ in the context of $Apc$ loss. Interestingly we found that the proliferative reduction seen in $villin$Cre$^{ERT2}$ $Apc;^{fl/fl}$ $Srsf1^{fl/+}$ was observed predominantly in the intestinal villus zone (Fig. 3a–b and S3f) and this was independent of variations in crypt-villus size or levels of apoptosis (Figures S3g and S3i-S3j). We investigated this proliferative reduction further by harvesting

villi from either $Apc^{fl/fl}$ or $Apc;^{fl/fl}$ $Srsf1^{fl/+}$ tissue and generated villi-derived 3D organoids in vitro. The clonogenic capacity of these purified $Apc;^{fl/fl}$ $Srsf1^{fl/+}$ epithelial cells was strongly impaired (Fig. 3c–e) demonstrating a persistence of this phenotype in a stromal-independent context. To determine in more detail whether $Srsf1$ depletion affected normal intestinal stem cell homeostasis we generated inducible $vil$-Cre-ERT2 WT and $Srsf1^{fl/+}$ mice carrying the $Lgr5GFP$-$CRE^{ERT2}$ transgene[28]. Following tamoxifen-induced gene deletion, we analysed the LGR5 + ISC population using GFP IHC to visualise LGR5-GFP expression and flow cytometry to determine the percentage of GFP positive cells. Both analyses showed that $Srsf1$ depletion had no impact on the LGR5 ISC population (Figure S3l–n). In addition, sorted single LGR5-GFP positive cells had the same capacity to form organoids when plated in vitro and showed no changes stem cell marker gene expression following $Srsf1$ depletion (Figure S3o–q). Together, these data indicate $Srsf1$ depletion does not affect normal LGR5 + intestinal stem cell homeostasis but significantly impairs the growth of $Apc$ deficient cells.

**Oncogenic SRSF1 levels affect cell-type plasticity.** To investigate whether the reduced proliferative phenotype observed in the intestinal mucosa of $Apc;^{fl/fl}$ $Srsf1^{fl/+}$ was due to changes in intestinal cell identity, we performed RNAseq analysis of $Apc^{fl/fl}$ and $Apc;^{fl/fl}$ $Srsf1^{fl/+}$ intestines. We compared the $Apc;^{fl/fl}$ $Srsf1^{fl/+}$ transcriptome (Supplementary Data 8) with previously defined signatures of intestinal cell types[29,30] using gene set enrichment analysis (GSEA). Interestingly, we found a significant overlap between genes associated with late transit-amplifying and differentiated enterocytes and gene overexpressed in $Apc;^{fl/fl}$ $Srsf1^{fl/+}$ intestinal cells (Figs. S3r–S3u) (Supplementary Data 9). Numerous differentiated cell marker genes were overexpressed in $Apc;^{fl/fl}$ $Srsf1^{fl/+}$ compared to $Apc^{fl/fl}$, including $Slc13a2$ and $Apoc2$ (Fig. 3f). We also found that expression of the Wnt target gene and cancer stem cell marker $Prox1$ was decreased in $Apc;^{fl/fl}$ $Srsf1^{fl/+}$ (Fig. 3g). Interestingly, the increase in the splicing iso-form ratio of $Kras4B/Kras4A$ observed after $Apc$ deletion (Fig. 1c and S1f) was reversed upon simultaneous deletion of $Srsf1$ (Fig. 3h). This was not due to alterations in nuclear localisation of β-catenin, a previously described function of Srsf1, as demonstrated by IHC analysis (Fig. S3v, w)[31]. These data suggest that $Srsf1$ promotes a less differentiated, stem cell-like phenotype following $Apc$ deletion. To investigate the generality of these findings we depleted $Srsf1$ using shRNA in a number of tumourigenic organoid models (Fig. S3x–z)[32,33] and determined stem cell function using clonogenicitiy assays. Validating our in vivo experiments, $Srsf1$ knockdown led to a significant reduction in clonogenicity, growth rate and viability of $Apc^{fl/fl}$ organoids suggesting reduced stem cell function (Fig. 3i–k and S3x–z). Notably, the same results were observed following $Srsf1$ knockdown in two mouse CRC organoid lines that model late-stage, metastatic disease (Fig. 3i–k and S3x–z). The expression of the stem cell marker $Lgr5$ was also reduced in $Srsf1$ depleted

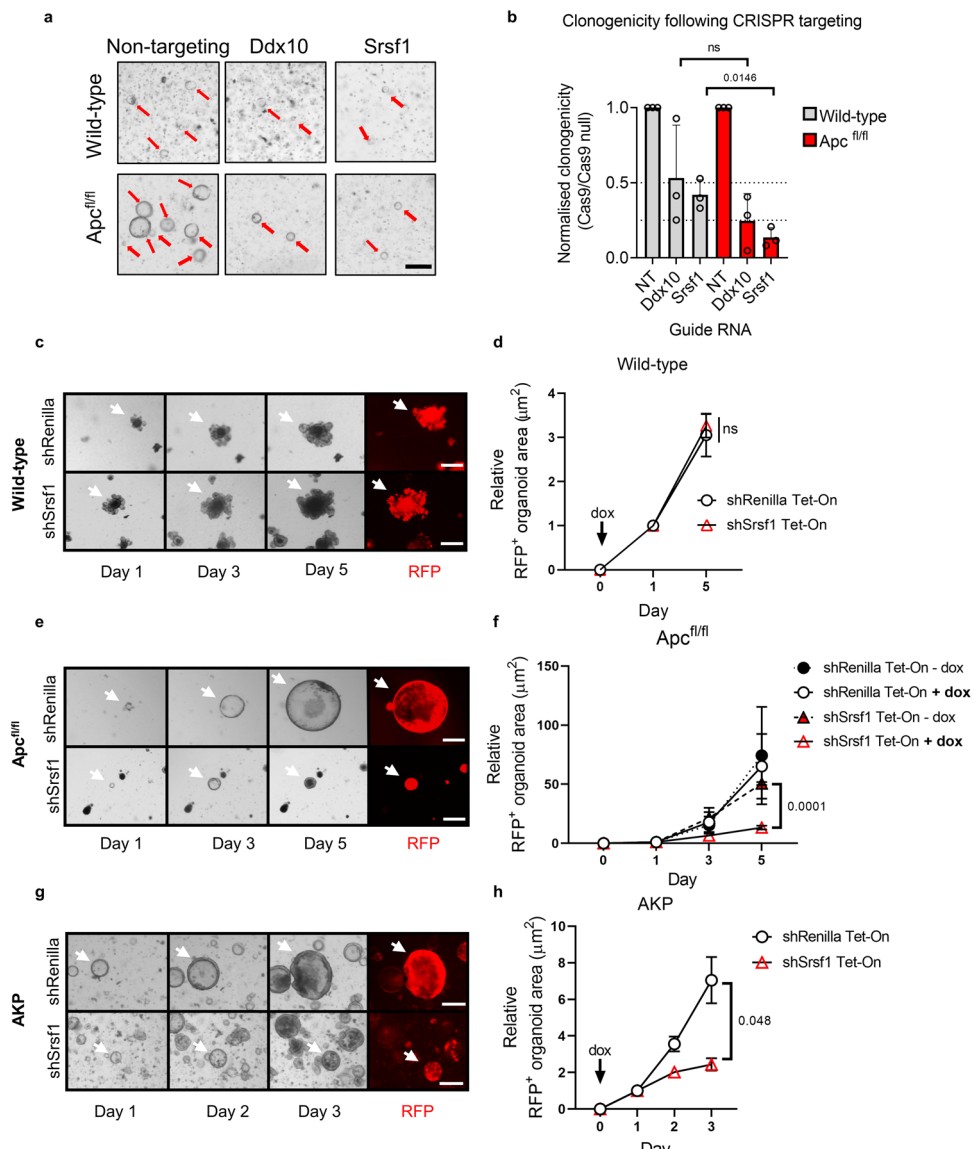

**Fig. 2 The splicing factor SRSF1 is critical for Apc-deficient growth. a** Images of clonogenicity assays in wild-type and *Apc*<sup>fl/fl</sup> Cas9 organoids, treated with the indicated gRNA. Scale bar 250 μm. **b** Clonogenicity quantification of organoids treated with indicated gRNA. **c** Images of doxycycline (dox)-induced (Tet-On) shRNAs against *Renilla* or *Srsf1* in wild-type intestinal organoids. Red fluorescent protein (RFP) is co-expressed with the shRNA following addition of dox. White arrows indicate a tracked organoid over the time-course following dox treatment. **d** Organoid size of tracked wild-type intestinal organoids. **e** Images of tracked *Apc*<sup>fl/fl</sup> organoids treated with the indicated Tet-On shRNA following addition of dox. **f** Size quantification of *Apc*<sup>fl/fl</sup> organoids with the indicated shRNA, +/− addition of dox. Statistical test indicated for shSrsf1 +/− dox. **g** AKP organoids with the indicated Tet-On shRNA following addition of dox. **h** Quantification of AKP organoid size. For (**c**, **e** and **g**) scale bar is 250 μm. All images shown are representative. Data shown as mean and error bars are SD (**b**) and SEM (**d**, **f**, **h**). All data are $n = 3$ independent experiments and analysed with two-tailed, unpaired $t$-tests, with (**d**, **f**, **h**) using Bonferroni post-hoc correction (alpha = 0.05), $p$ values are indicated in figure panels. See also Fig. S2.

organoid lines (Fig. S2d). Together, these data demonstrate that *Srsf1* depletion reduces stem cell properties and results in colorectal cancer cells adopting a more differentiated cell transcriptional phenotype.

**SRSF1 facilitates intestinal cell dedifferentiation**. Due to these findings we hypothesised that SRSF1 is required for differentiated enterocytes to dedifferentiate and acquire stem cell properties. Activation of KRAS or IκB kinase (IKK) combined with *Apc* loss has been shown to induce dedifferentiation of villus enterocytes[34]. Therefore, we generated mice carrying *villin*Cre<sup>ERT2</sup> *Apc*;<sup>fl/fl</sup> *Kras*<sup>G12D</sup> or *villin*Cre<sup>ERT2</sup> *Apc*;<sup>fl/fl</sup> *Kras*;<sup>G12D</sup> *Srsf1*<sup>fl/+</sup> alleles and induced gene recombination with tamoxifen. As previously

described[34,35], a shorter 3-day post tamoxifen time point was employed so that dedifferentiating cells in the villus could be studied alone, without contamination from recombined cells migrating from the crypts. After 3-days, villi from the small intestines were dissected, digested to single cells and plated to determine dedifferentiation efficiency (Fig. S4a). We compared enterocyte dedifferentiation between *villin*Cre<sup>ERT2</sup> *Apc*;<sup>fl/fl</sup> *Kras*<sup>G12D</sup> and *villin*Cre<sup>ERT2</sup> *Apc*;<sup>fl/fl</sup> *Kras*;<sup>G12D</sup> *Srsf1*<sup>fl/+</sup> mice and found that depletion of *Srsf1* led to a significant reduction in the proportion of organoids formed (Fig. 4a, b). The clonogenic capacity of the organoid lines derived from these dedifferentiated cells was also reduced in the *Srsf1*<sup>fl/+</sup> genotype (Fig. 4c) suggesting that the acquisition of stem cell properties was impaired. Intriguingly, *Kras* isoform splicing was still modulated by SRSF1

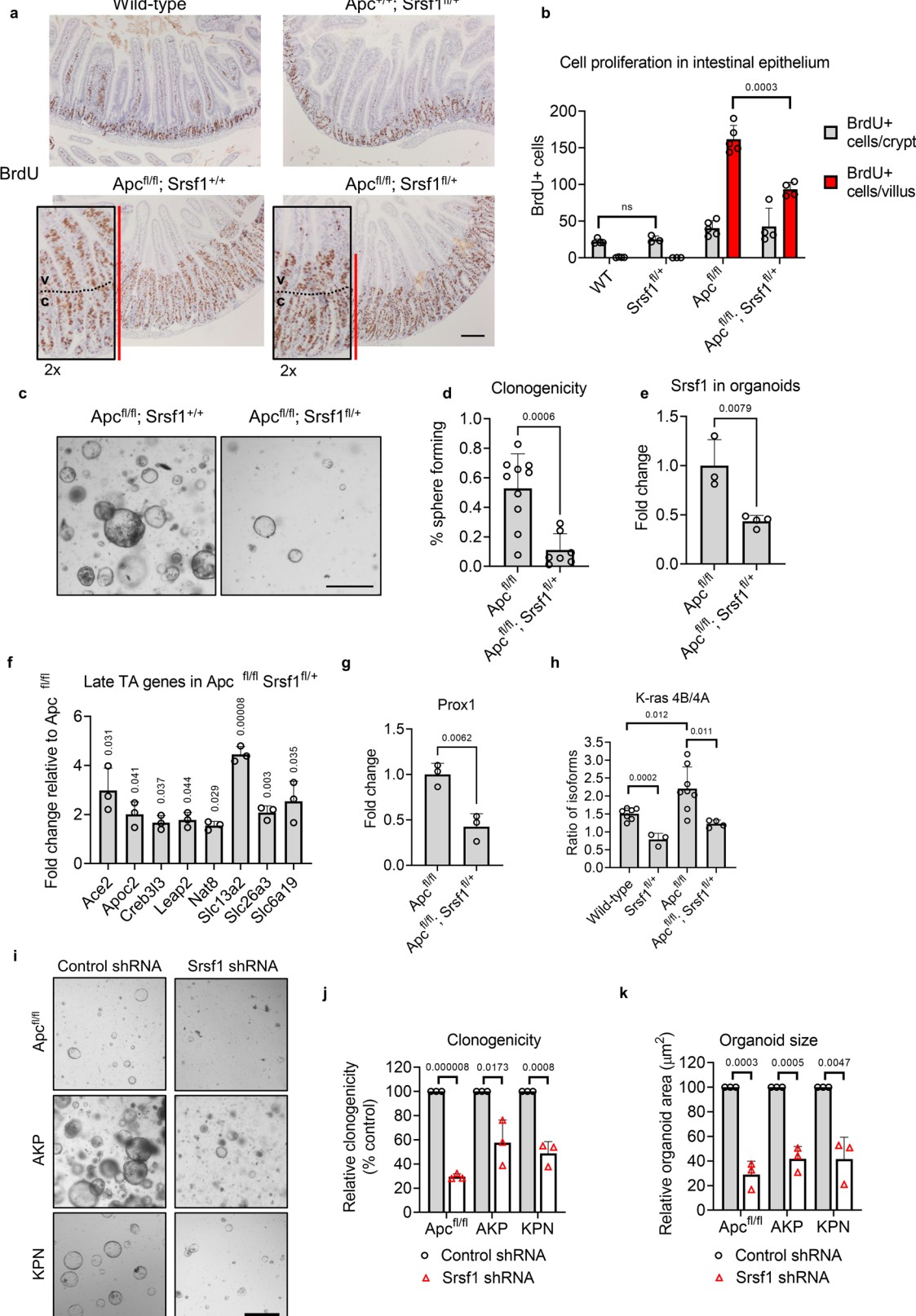

even in the context of constitutively active KRAS signalling (Figure S4b).

To confirm these observations, we employed a different dedifferentiation model driven by hyperactive NF-κB signalling[34], (*villin*Cre[ERT2] *Apc*[fl/fl] *IKK2*[ca]) and subjected these mice to the same experimental procedure. Again, we found a similar

impairment in dedifferentiation capacity of villi cells and reduced clonogenicity in derived organoids following heterozygous *Srsf1* deletion (Fig. 4d–f).

To rule out the possibility that these clonogenic cells derived from crypt cells migrating into the villi we sorted differentiated villus cells based on expression of EPHB2. EPHB2 marks a

**Fig. 3 SRSF1 is required for epithelial cell hyperproliferation and affects cell-type plasticity. a** Representative images of BrdU-stained mouse small intestines with the indicated genotypes, 5-days after tamoxifen induction. Letters 'c' and 'v' indicate demarcated crypt and villus compartments respectively and the red line highlights the proliferative zone. Scale bar 250 µm. **b** Quantification of proliferative cells in the crypts and villi of the indicated genotypes, $n = 4$ (WT) vs 3 (Srsf1$^{fl/+}$) vs 5 (Apc$^{fl/fl}$) vs 4 (Apc;$^{fl/fl}$ Srsf1$^{fl/+}$) biologically independent mice. **c** Representative images of clonogenicity assays using villi-derived intestinal organoids of indicated genotypes. Scale bar 500 µm. **d** Quantification of villi-derived organoid clonogenicity, $n = 10$ vs 7 independent experiments. **e** qPCR gene expression of Srsf1 in organoids normalised to β-actin, $n = 3$ vs 4 independent experiments. **f** qPCR quantification of upregulated late transit-amplifying (differentiated) cell markers in Apc;$^{fl/fl}$ Srsf1$^{fl/+}$ mouse intestinal tissue, relative to Apc$^{fl/fl}$ intestinal tissue, using Gapdh loading control, $n = 3$ vs 3 biologically independent mice. **g** Prox1 stem cell marker qPCR quantification in mouse small intestines of indicated genotypes, normalised to Gapdh, $n = 3$ vs 3 biologically independent mice. **h** qPCR-derived ratio of Kras splicing isoforms in the indicated genotypes, $n = 7$ (WT) vs 3 (Srsf1$^{fl/+}$) vs 8 (Apc$^{fl/fl}$) vs 4 (Apc;$^{fl/fl}$ Srsf1$^{fl/+}$) biologically independent mice. **i** Images of organoids of the indicated genotype, treated with control or Srsf1 shRNA. AKP (Apc;$^{\Delta/\Delta}$ Kras;$^{G12D/+}$ Trp53$^{\Delta/\Delta}$), KPN (Kras;$^{G12D/+}$ Trp53;$^{fl/fl}$ Rosa26$^{N1icd/+}$). Scale bar 500 µm. **j** Relative clonogenicity and **k** size of surviving organoids following control or Srsf1 shRNA treatment, $n = 3$ vs 3 independent experiments. Data in bar charts are represented as mean and error bars are SD with data analysed with two-tailed, unpaired t-tests, p values are indicated in figure panels. All biological replicates are shown as individual value plots. See also Fig. S3.

gradient of stem and progenitor crypt cells with expression absent in differentiated villus cells[30]. We sorted EPHB2 negative epithelial cells from wild-type crypts and confirmed that they did not have clonogenic capacity (Figure S4c–e). We next utilised this sorting strategy to obtain differentiated villus cells from induced villinCre$^{ERT2}$ Apc;$^{fl/fl}$ Kras$^{G12D}$ and villinCre$^{ERT2}$ Apc;$^{fl/fl}$ Kras;$^{G12D}$ Srsf1$^{fl/+}$ mice (Fig. 4g). EPHB2 negative differentiated Apc;$^{fl/fl}$ Kras$^{G12D}$ cells were able to form colonies, indicative of cellular dedifferentiation. This ability was significantly impaired following Srsf1 depletion, demonstrating the requirement of Srsf1 for the acquisition of stem cell properties (Fig. 4h, i). To confirm these findings we utilised an analogous model where doxycycline-inducible shRNA depletion of Apc is used to activate Wnt signalling in Kras$^{G12D}$ mutant colonic organoids[36]. In the absence of doxycycline, organoid growth is dependent on media supplementation with Wnt3a and R-spondin. In addition, EPHB2 negative cells have poor clonogenic capacity indicative of a differentiated cell phenotype (Figure S4f). We used shRNA to deplete Srsf1 in these organoids and following doxycycline induction FACS sorted for EPHB2 negative, differentiated cells. Again, depletion of Srsf1 significantly impaired the ability of such cells to form colonies (Fig. 4j–l). In addition, organoids derived from Srsf1 depleted differentiated cells expressed significantly higher levels of the differentiated cell markers Muc2 and Krt20 (Figure S4g, S4h). Together, these data demonstrate a requirement for Srsf1 expression in mediating intestinal cell plasticity.

To determine whether Srsf1 also drives increased stem cell properties, we overexpressed wild-type Srsf1 (Srsf1$^{WT}$) or Srsf1 with mutations in the second RNA recognition motif (RRM2) leading to impaired RNA binding[37], (Srsf1$^{D136A,K138A}$) in Apc$^{fl/fl}$ organoids (Figure S4i). Consistent with our deletion studies, Srsf1$^{WT}$ overexpression led to increased organoid growth, clonogenicity and increased expression of the stem cell marker Lgr5 (Figures S4i–l). By contrast the Srsf1 mutant did not affect organoid growth, demonstrating that the RNA-binding ability of SRSF1 is necessary for its dedifferentiation activity. Based on these findings, we conclude that SRFS1 mediates intestinal cell dedifferentiation and the acquisition of stem cell properties.

**Wnt-induced SRSF1 levels enforce splicing dysregulation.** To ascertain the extent to which Srsf1 contributes to Wnt-driven splicing dysregulation and identify potential splicing changes that mediate oncogenesis, we conducted differential splicing analysis on Apc$^{fl/fl}$ and Apc;$^{fl/fl}$ Srsf1$^{fl/+}$ intestinal tissue, in which we observed reduced proliferation and stem cell activity, using SUPPA2. We identified 577 alternative splicing events ($p < 0.05$) (Fig. 5a and Supplementary Data 10). The majority of these were exon skipping alternative splicing events (30.5%), supporting

previous work using in vitro models showing that SR proteins promote splice site selection through exonic splicing enhancer recognition[38]. We then determined the proportion of unique dysregulated alternative splicing events in Apc$^{fl/fl}$ that were SRSF1-dependent. We compared the alternative splicing events identified from our wild-type/Apc$^{fl/fl}$ analysis (Fig. 1b and Supplementary Data 3) with the alternative splicing events found in Apc$^{fl/fl}$/Apc;$^{fl/fl}$ Srsf1$^{fl/+}$ (Supplementary Data 10). There was a significant enrichment of alternative splicing events from Apc$^{fl/fl}$/Apc;$^{fl/fl}$ Srsf1$^{fl/+}$ that were also found in our wild-type/Apc$^{fl/fl}$ dataset ($p < 1e^{-5}$) (Fig. 5b). In addition, a significant proportion of the alternative splicing events occurring after Apc deletion were reverted upon deletion of Srsf1$^{fl/+}$ (127 alternative splicing events with discordant dPSIs, $p = 0.049$) (Supplementary Data 11). Again, rMATS analysis identified a large proportion of potentially novel splicing events suggesting widespread changes in alternative splicing mediated by SRSF1 (Figure S5a and Supplementary Data 12).

We validated several of the alternative splicing events discovered from our differential splicing analysis (Fig. 5c, d). Additionally, isoform level changes in Kras splicing were detected and validated confirming that the Wnt-induced increase in the pro-proliferative Kras4b isoform over the pro-apoptotic Kras4a isoform was reversed upon Srsf1 depletion. SRSF1-dependent changes in Kras isoform splicing levels were also present in multiple other models (Figs. 3h, S4b, S6j and S6m). To determine whether these transcripts were direct targets of SRSF1 we carried out RNA immunoprecipitation experiments. By pulling down SRSF1 from the CMT93 mouse colorectal cancer cells line and carrying out qRT-PCR we identified significant binding of Srsf1 to several of the alternatively spliced transcripts identified by RNAseq (Fig. S5b, c). In addition, alternative splicing events (events identified via SUPPA2 analysis, not including AF events) in our Apc$^{fl/fl}$/Apc;$^{fl/fl}$ Srsf1$^{fl/+}$ dataset, which were repressed following Srsf1 depletion, were enriched for SRSF1 binding motifs, together suggesting loss of direct, functional SRSF1 binding to these transcripts led to the splicing changes we observed (Fig. S5d). Besides the well-characterised role of SRSF1 in pre-mRNA splicing in the nucleus, this shuttling SR protein also has post-splicing functions, which include mRNA export, mRNA translation and nonsense-mediated decay[12,39]. We utilised a novel mouse model carrying a nucleo-cytoplasmic shuttling defective Srsf1 mutant to determine whether non-nuclear functions of SRSF1 could explain the phenotypes we observed[40]. Apc deficient organoids carrying a single allele of Srsf1-NRS (which is unable to shuttle to the cytoplasm) had no defect in clonogenic capacity (Fig. S5e–g). This is unlike organoids carrying heterozygous Srsf1 deletion (Fig. 3c, e) demonstrating that the cytoplasmic function of Srsf1 does not

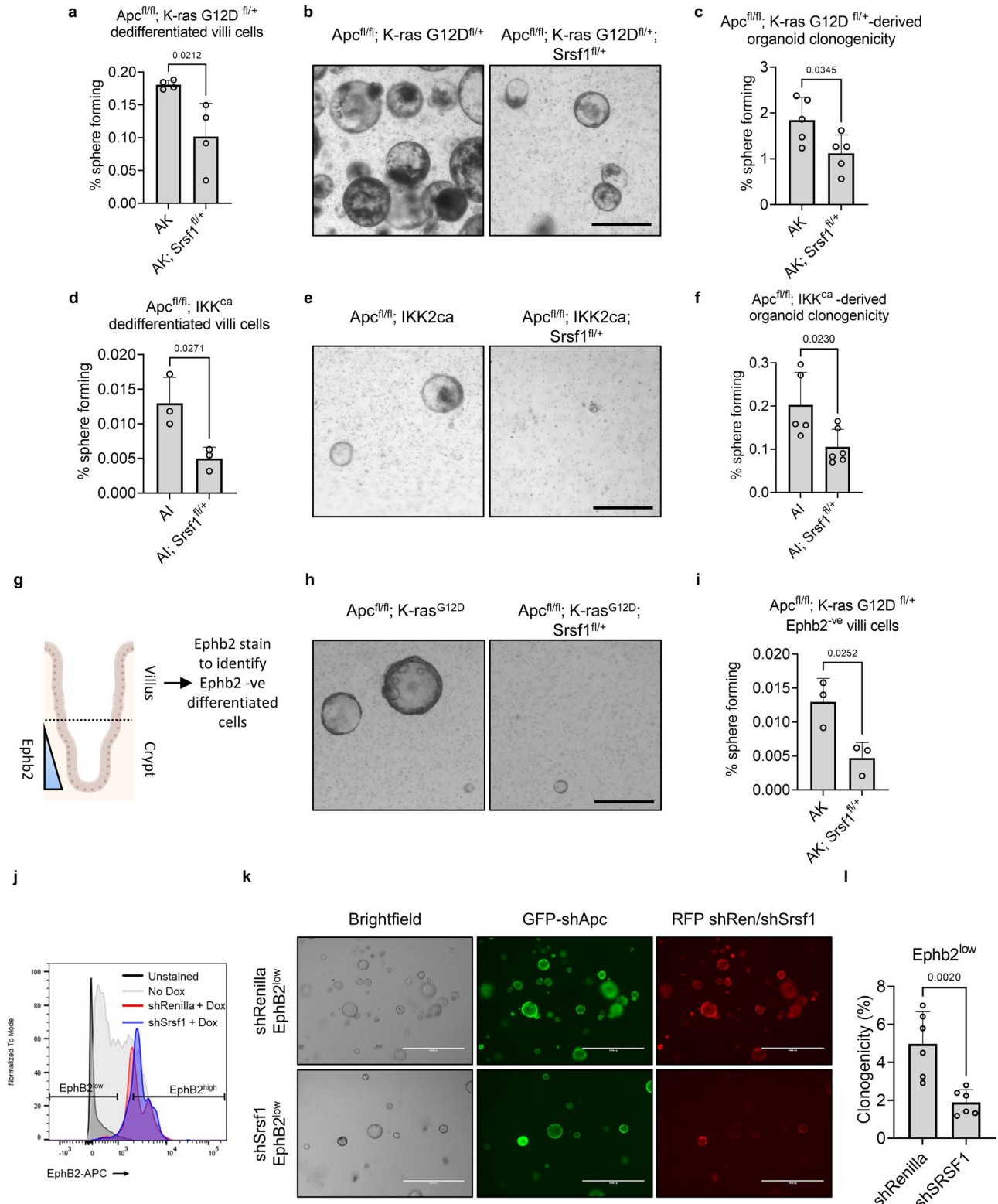

explain the phenotypes we observe in this model. Together, these data suggest the primary cause of the effects we observed are due to nuclear (presumably splicing) functions of SRSF1.

To functionally determine whether *Kras4b* plays a role in colorectal tumourigenesis we treated *Apc*[fl/fl] and metastatic KPN (*Kras*;[G12D/+] *Trp53*;[fl/fl] *Rosa26*[N1icd/+]) organoids with the KRAS4B specific inhibitor deltarasin (Fig. S5h)[41,42]. Deltarasin is a small molecule inhibitor of the KRAS -PDEδ interaction,

and PDEδ has been shown to chaperone and recruit KRAS4B to the plasma membrane, but is not required for KRAS4A recruitment. Treatment with deltarasin led to a rapid organoid dissociation and a highly significant reduction in viability of both *Apc*[fl/fl] and KPN organoid lines demonstrating an important role for KRAS4B in maintaining CRC organoid viability (Fig. 5e, f). Treatment of wild-type organoids led to a smaller reduction in viability (Fig. 5e, f). Treatment of human normal and CRC

**Fig. 4 SRSF1 facilitates intestinal cell dedifferentiation. a** Dedifferentiation assay using AK ($Apc;$<sup>fl/fl</sup> $Kras$<sup>G12D</sup>) cells directly purified from mouse intestinal villi from the indicated genotypes isolated 3 days postinduction, and quantification of their ability to form organoid clones, $n = 4$ vs 4 biologically independent mice. **b** Images from AK dedifferentiation assays. **c** Clonogenicity assays on organoids derived from and passaged from dedifferentiated villi-derived AK cells, $n = 5$ vs 5 independent experiments. **d** An alternative dedifferentiation assay using AI ($Apc;$<sup>fl/fl</sup> $IKK2$<sup>ca</sup>) cells directly purified from mouse intestinal villi from the indicated genotypes, and quantification of their ability to form organoid clones, $n = 3$ vs 3 biologically independent mice. **e** Images from AI dedifferentiation assays. **f** Clonogenicity assays on organoids derived from and passaged from dedifferentiated villi-derived AI cells, $n = 5$ vs 6 independent experiments. **g** Schematic of experimental strategy to isolate differentiated villus cells 2 days post tamoxifen induction. **h** Images from EPHB2 –ve cell sorted AK dedifferentiation assays. **i** Quantification of colony-forming capacity of differentiated cells, $n = 3$ vs 3 biologically independent mice. **j** Overview of FACs plots used to gate for EPHB2<sup>high</sup> and EPHB2<sup>low</sup> cells. **k** Representative images of colonies formed from EPHB2<sup>low</sup> cells. GFP indicates $Apc$ knockdown and RFP indicates $Renilla$ control or $Srsf1$ knockdown. **l** Quantification of colony formation, $n = 6$ vs 6 independent experiments. Scale bars in **b**, **e** and **h** are 500 μm and in K 1000 μm. All images are representative. Data in bar charts are represented as mean and error bars are SD with data analysed with two-tailed, unpaired t-tests, $p$ values are indicated in figure panels. All biological replicates are shown as individual value plots. See also Fig. S4.

organoids showed a similar effect, with organoids derived from normal colon being resistant but those derived from tumours being highly sensitive to deltarasin treatment (Fig. S5i, S5j). To validate these findings we designed an antisense morpholino that inhibits the splicing of $Kras4b$. Treatment of organoids with this morpholino proved highly effective at reducing the $Kras4b/Kras4a$ splice ratio (Fig. S5m, S5n) and significantly reduced the clonogenic capacity and viability of $Apc$<sup>fl/fl</sup> and KPN organoids (Fig. S5k–S5p).

To further this analysis, we next asked whether $Kras4b$ expression is sufficient to rescue organoid viability in $Srsf1$ depleted organoids. We generated organoid lines overexpressing GFP alone, or $Kras4b$ (also marked with GFP expression) both carrying a DOX inducible $shSrsf1$ RFP construct. In this model $Kras4b$ expression (marked by GFP) can be maintained alongside $Srsf1$ depletion (marked by RFP expression) (Fig. S5q). We allowed organoids to form, induced with DOX and tracked GFP/RFP double-positive organoids over 3 days. Similar to our previous results, depletion of $Srsf1$ in GFP expressing controls led to a significant reduction in organoid growth (Fig. 5g, h). However, this growth reduction was significantly rescued to wild-type levels by ectopic expression of $Kras4b$ indicative of a functional role for $Kras4b$ in mediating the phenotypic effects of SRSF1 expression in this model (Fig. 5g, h).

To infer the biological function of the two $Kras$ splicing isoforms in the context of CRC, we employed proximity-dependent biotin identification (BioID) to uncover interacting proteins of KRAS4A and KRAS4B (Fig. S5r–S5u). We found 83 proteins that significantly ($p < 0.05$) interacted with both KRAS4A and KRAS4B (Fig. S5t and Supplementary Data 13). We also identified 23 proteins that monogamously interacted with KRAS4A and 22 proteins that uniquely interacted with KRAS4B. Some of these uniquely interacting proteins were associated with contrasting cell signalling pathways and associated with unique cellular compartments (Supplementary Data 13). For example, IGF1R and RALA interacted only with KRAS4A whereas BRAF and RAP1A were exclusively associated with KRAS4B. These data show that KRAS4A and KRAS4B have specific subsets of protein interactors indicating SRSF1-controlled splicing changes can result in significant changes in the oncogenic protein interactome.

**High $Srsf1$ expression mediates tumour cell plasticity and colorectal cancer invasiveness.** To explore the role of SRSF1 in advanced stage colonic carcinogenesis we utilised a previously described model of carcinogen-induced tumourigenesis[43]. Cohorts of $villin$Cre<sup>ERT2</sup> $p53$<sup>fl/fl</sup> and $villin$Cre<sup>ERT2</sup> $p53;$<sup>fl/fl</sup> $Srsf1$<sup>fl/+</sup> mice were treated with repeated rounds of azoxymethane (AOM) and aged until signs of colonic tumourigenesis became apparent (Fig. S6a). Mice treated with AOM presented predominantly with colonic

tumours, with tumours of the small intestine rarely occurring (Fig. S6b). Although we did not find evidence that targeting $Srsf1$ provided a survival advantage or a change in tumour number or burden (Figs. S6c–S6e) we found that deletion of one copy of $Srsf1$ (Fig. S6f) significantly reduced the proportion of mice presenting with invasive tumours in this model (Fig. 6a, b). Additionally, overall number and percentage of invasive tumours per mouse and the presence of extensive collagen deposition was significantly reduced in $Srsf1$ depleted mice (Figs. 6c and S6g–S6i).

Upon examination of the invasive tumours, we found evidence of cell plasticity changes. Immunohistochemical analysis of the cancer stem cell marker PROX1 revealed a significant decrease in PROX1 expression in invasive $p53;$<sup>fl/fl</sup> $Srsf1$<sup>fl/+</sup> tumour epithelial cells (Fig. 6d, e) corroborating our RNAseq results in short-term Cre-lox recombined $Apc;$<sup>fl/fl</sup> $Srsf1$<sup>fl/+</sup> tissue (Fig. 3g). We also found a significant corresponding increase in expression in the differentiation marker, SLC13A2 (Figs. 6d, f), and altered $Kras$ splicing in these tumours (Fig. S6j). These data suggest that impaired $Srsf1$ expression reduces the invasive and stem cell behaviour of colonic tumours in vivo.

To determine whether SRSF1 directly controls these phenotypes, we cultured 3D tumour organoids from invasive AOM $p53$<sup>fl/fl</sup> tumours and depleted $Srsf1$ expression using shRNA (Fig. S6k). We found that both PROX1 expression and $Kras4b/Kras4a$ splicing ratio were impaired upon $Srsf1$ knockdown confirming a direct role for $Srsf1$ in mediating these effects (Figs. S6l, S6m). We then carried out an organoid invasion assay and observed a significant reduction in organoid cell invasion through matrigel upon $Srsf1$ depletion (Fig. 6g–i).

Together, these data suggest that SRSF1 directly controls tumour cell invasion and maintenance of stem cell properties in late-stage colon cancer.

**$SRSF1$ depletion inhibits growth and stemness of human colorectal cancer-derived organoids.** To ascertain how SRSF1 levels correlate with human CRC progression, we analysed colon cancer tissue microarrays (TMAs) containing varying severities of the disease. First, we investigated if the cancer stem cell marker PROX1, which has been show to promote metastatic outgrowth of cells[44], correlated with SRSF1 expression. We found a positive correlation of SRSF1 and PROX1 expression (Fig. 7a, b and S7a) similar to that seen in our laboratory models (Figs. 3g and 6d, e). We next examined SRSF1 expression in different tumour stages and found that more invasive and metastatic tumours had significantly higher expression of SRSF1 (Fig. 7c) showing that SRSF1 levels correlate with advanced-stage colorectal cancer.

To phenocopy clinical intervention by way of attenuating $SRSF1$ levels in human CRC, we manipulated $SRSF1$ in patient-

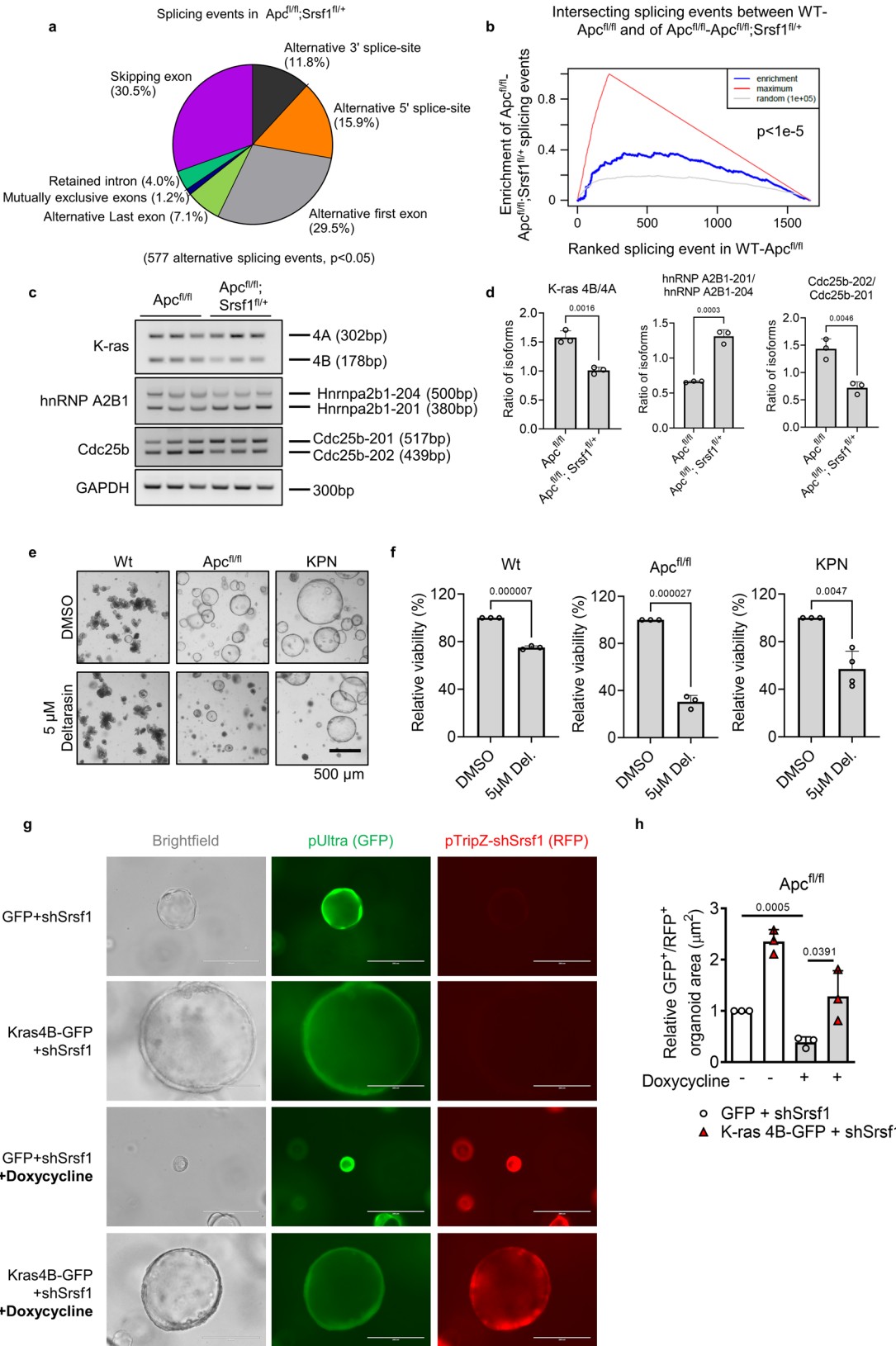

derived organoids (PDOs) using shRNA. Consistent with our animal experiments the clonogenicity, size and viability of the PDOs was impaired by depleting *SRSF1* expression (Fig. 7d–g). Reduced *SRSF1* expression was associated with a corresponding reduction in *LGR5* suggesting that in human cancer, as in our mouse models, SRSF1 mediates stem cell function (Figs. 7h and

S7b–S7d). We also observed an impairment in the ratio of *KRAS4B/KRAS4A* (Fig. 7h) which we had earlier found to be critical in facilitating oncogenic growth in mouse (Fig. 5e, f).

Taken together these data establish that, not only can SRSF1 attenuation mitigate the oncogenic potential of CRC in mouse models, but also in the human disease.

**Fig. 5 Wnt-induced SRSF1 levels enforce splicing dysregulation. a** Alternative splicing events in Apc;[fl/fl] Srsf1[fl/+] compared to Apc[fl/fl] in small intestine isolated 5 days postinduction. **b** Gene set enrichment analysis showing the intersecting alternative splicing events discovered in the wild-type versus Apc;[fl/fl] Srsf1[fl/+] dataset compared to the Apc[fl/fl] versus Apc;[fl/fl] Srsf1[fl/+] dataset. **c** RT-PCR validation of a selection of alternative splicing events and **d** quantification, n = 3 vs 3 biologically independent mice. **e** Images of wild-type, Apc[fl/fl] and KPN organoids treated with deltarasin or control vehicle DMSO. Scale bar 500 μm. **f** Relative cell viability following deltarasin treatment in wild-type, Apc[fl/fl] and KPN organoids, n = 3 vs 3 (Wt and Apc[fl/fl]) and 3 vs 4 (KPN) independent experiments. **g** Images showing Apc[fl/fl] organoids following treatment with a doxycycline (dox) inducible Srsrf1 shRNA +/− dox, and with GFP or Kras4b-GFP overexpression. RFP is co-expressed with the shRNA following addition of dox. Scale bar 500 μm. **h** Quantification of organoid size under the stated conditions, n = 3 vs 3 vs 3 independent experiments. All images are representative. Data in bar charts are represented as mean and error bars are SD with data analysed with two-tailed, unpaired t-tests, p values are indicated in figure panels. All biological replicates are shown as individual value plots. See also Fig. S5.

## Discussion

Tumour growth has been shown to increase the transcriptional output of cells leading to an elevated burden of pre-mRNAs requiring processing, including an increased requirement for RNA splicing[45]. The RNA addiction of hyperproliferative cells might therefore be exploited for clinical purposes. This idea was supported by earlier work showing that the spliceosome is a potential therapeutic vulnerability following MYC induction in 2D breast cancer cells and also during lymphomagenesis[18,19]. Here we have shown that genes associated with RNA metabolic processing are upregulated in a mouse model of colorectal cancer initiation. These include genes involved in RNA splicing. 60 splicing related genes are upregulated following Apc loss, accounting for approximately a quarter of all known splicing genes. Based on this, we surmised that splicing impairment might effectively target rapidly proliferating intestinal organoids with hyperactive Wnt signalling. Indeed, treating organoids with pladienolide B (which targets the SF3B complex) successfully targeted Apc[fl/fl] organoids without adversely affecting wild type organoids suggesting that RNA splicing generally may act as a therapeutic vulnerability in CRC. However, this drug has recently failed in clinical trials as it caused optic-nerve dysfunction and vision loss in patients[46]. On the other hand, a novel drug targeting the splicing factor SRSF1, ABX300, has been shown to impair SRSF1 splicing activity to treat diet-induced obesity in mice without any observed toxicity[47]. Furthermore, villinCre[ERT2] Srsf1[fl/+] mice had normal intestinal morphology without any change in cell proliferation or normal stem cell function, and an inducible shRNA against Srsf1 had no observable phenotype in wild-type organoids. Thus, there is a clear therapeutic window for targeting SRSF1 in colorectal cancer.

In conjunction with an increased requirement for splicing factors, we identified dysregulation of splicing itself, as shown by the shift in RNA isoforms generated following Apc loss. We found that the Kras4a and 4b isoforms were alternatively spliced depending on the level of Wnt activation, and this was mediated by SRSF1. SRSF1 has previously been proposed to be oncogenic and is a direct transcriptional target of Myc[48,49]. In addition, SRSF1 expression has been shown to be dependent on Wnt signalling in colorectal cancer suggesting that Wnt driven dysregulation of RNA splicing is partially mediated by SRSF1 following Apc loss[50]. The Kras4a isoform has been shown to promote cell death[24] and, when mutated, has significantly less oncogenic potential in mice than Kras4b[51]. Not only did we see a shift in splicing of these isoforms following Apc deletion, but we found that when we impaired the function of Apc-upregulated KRAS4B using deltarasin or an antisense morpholino, we could hamper oncogenic growth in our organoid models. Furthermore, the expression of Kras4b could rescue the growth of cells with impaired Srsf1 levels. As Srsf1 depletion leads to alterations in numerous splicing events it is unlikely that this single splicing event fully explains the role of SRSF1 in CRC. It is likely that numerous SRSF1 targets contribute to its phenotypic effect,

making SRSF1 an attractive therapeutic target, depletion of which could impact on multiple different pathways.

Targeting Srsf1 resulted in reduced proliferation and stem cell function consistent with similarly described functions of SRSF1 in breast cancer[50,52]. Transcriptome analysis of Apc;[fl/fl] Srsf1[fl/+] intestinal tissue revealed a role for SRSF1 in modulating cellular plasticity with the gene expression signature becoming distinctly enterocyte-like. Dedifferentiation of enterocytes has been shown to support a 'top-down' model of colorectal tumour morphogenesis where adenomas originate at the top of intestinal crypts[34,53]. Our data support a role for SRSF1-driven dedifferentiation in enterocytes. Following early activation of constitutive KRAS or NF-kB signalling with Apc loss, a reduction in dedifferentiation-driven clonal events arising from villi-derived differentiated cells was observed with impaired Srsf1 levels. Thus, SRSF1 can mediate cellular plasticity. As SRSF1 is a splicing factor, such plasticity changes might be brought about indirectly as a result of a general splicing repression response, or due to a change in the RNA isoform repertoire. As well as showing increased expression of differentiation marker genes, Srsf1 reduction resulted in a decrease in the stem cell marker Prox1. PROX1 is a Wnt-regulated transcription factor that has been shown to advance colon cancer progression by promoting dysplasia in colonic adenomas[54], as well as enhancing metastasis in Wnt-driven progenitor cells[44]. Not only were Prox1 levels impaired in the intestines of Apc;[fl/fl] Srsf1[fl/+] mice following acute Apc deletion, but this reduced level was observed in the advanced tumours of AOM treated p53;[fl/fl] Srsf1[fl/+] mice. These tumours were also significantly less invasive and displayed evidence of the same cell differentiation phenotypes observed in our early-stage tumour model. Encouragingly, this demonstrates a potential to target SRSF1 mediated cellular plasticity even in the advanced stages of the disease.

In conclusion, our investigation using mouse models, ex-vivo organoid systems and patient-derived samples demonstrates that intestinal Wnt-driven cancers are addicted to the spliceosome. We have shown that targeting an individual oncogenic splicing factor, SRSF1, impairs cancer progression through a variety of mechanisms (Fig. 7i). Modulation of cell-type plasticity in favour of a gene expression signature with reduced stemness, an impaired ability of cells to dedifferentiate, as well as tumours having a lower invasive potential, all make targeting SRSF1 a highly attractive option, and may complement current standard-of-care therapies.

## Methods

**Contact for reagent and resources sharing**. Requests for further information, reagents and resources should be directed to and will be fulfilled by the Lead Contact, Kevin B. Myant: (kevin.myant@igmm.ed.ac.uk).

**Animals models**. Species used: Mus musculus. All animal experiments were performed in accordance with a UK Home Office licence (Project License 70/8885), and were subject to review by the animal welfare and ethics board of the University of Edinburgh. Both genders of mice were used for all experiments at an age of

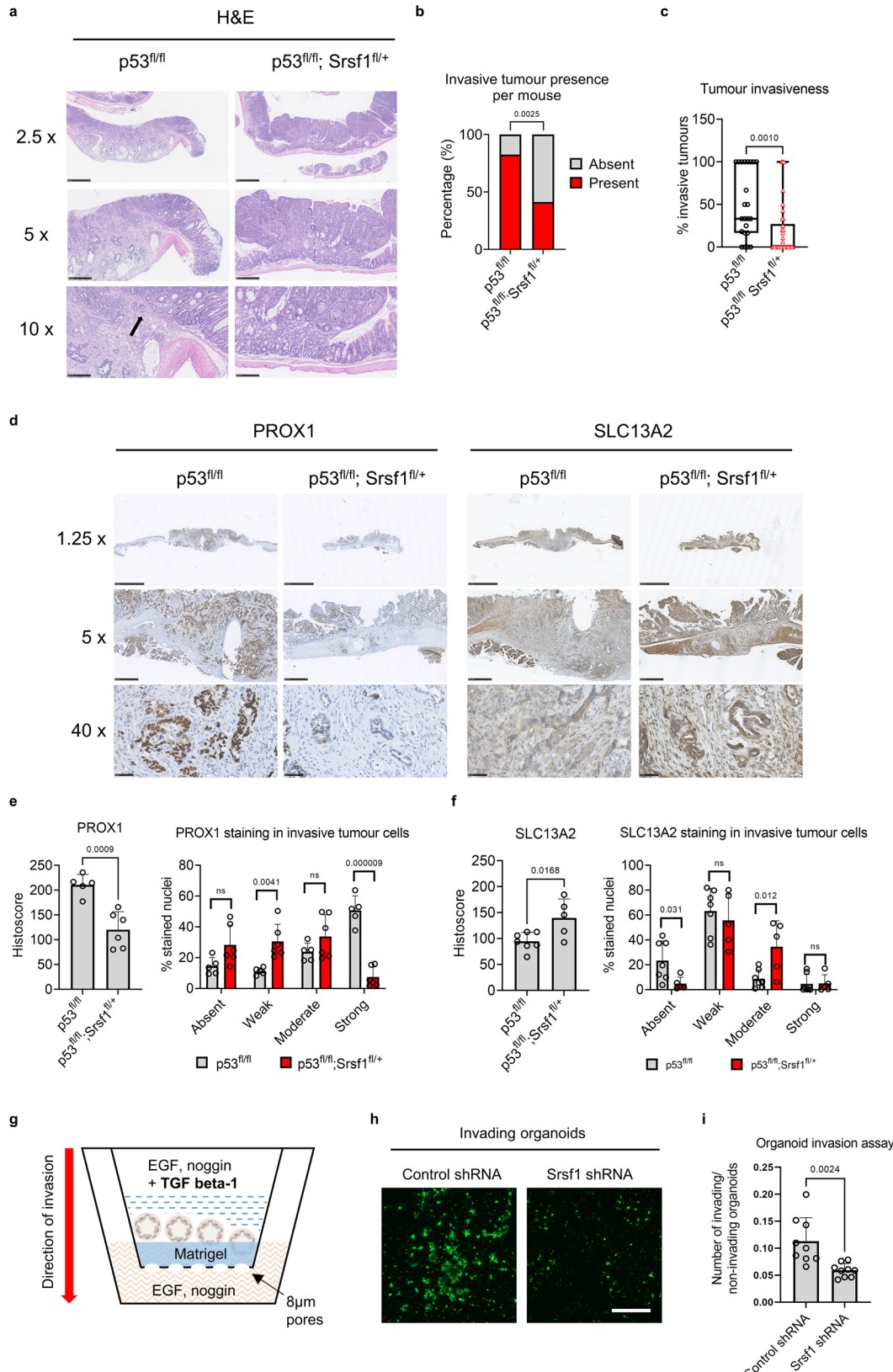

between 6 and 12 weeks once they had reached a minimum weight of 20 g. Mice were bred at the animal facilities of the University of Edinburgh and were kept in 12 h light–dark cycles and were given access to water and food ad libitum. Mice were maintained in a temperature- (20–26 °C) and humidity- (30–70%) controlled environment. Colonies had a mixed background (50% C57Bl6J, 50% S129). The genetic alleles used for this study were as follows: *villinCreER*[55], *Apc* (floxed)[56], *ASF/SF2* (Srsf1 floxed)[27], *Kras* (G12D)[57], *IKK2ca*[58], *P53* (floxed)[59]. Mice were

genotyped by Transnetyx (Cordoba, USA). At experiment endpoints, mice were humanely sacrificed by cervical dislocation (CD) in line with UK Home Office regulations.

*Tumour models and treatments.* For short-term Cre-Lox recombination where animals were taken at a specific time point of between 2 and 5 days postinduction

**Fig. 6 Srsf1 mediates tumour cell plasticity and colorectal cancer invasiveness. a** Representative histological images of mouse colons and tumours stained with hematoxylin and eosin (H&E). Scale bars are 1 mm (2.5x), 500 μm (5x) and 250 μm (10x). Tumours were isolated when mice showed clinical signs of intestinal tumourigenesis. **b** Quantification of the presence or absence of invasive tumours in mice from the indicated genotypes, with statistical difference calculated using a two-sided Chi-square test, $n = 23$ vs 34 biologically independent mice. **c** Box and whisker plot showing the number of invasive tumours as a percentage of the total number of tumours for each mouse for each genotype. The box extends from 25th to 75th centiles, the centre line is median and whiskers extend to minima and maxima, $n = 23$ vs 34 biologically independent mice. **d** Representative histological images of $p53^{fl/fl}$ and $p53^{fl/fl}$ $Srsf1^{fl/+}$ mouse intestines and tumours stained for PROX1 (stem cell marker) and SLC13A2 (differentiation marker). Scale bars are 2.5 mm (1.25x), 500 μm (5x) and 50 μm (40x). **e** Histoscore and staining strength quantification for PROX1, $n = 5$ vs 6 biologically independent mice and **f** SLC13A2, $n = 7$ vs 5 biologically independent mice. **g** Schematic depiction of the experimental strategy used to investigate the invasive potential of tumour-derived intestinal organoids. **h** Representative images of calcein-stained organoids once they had invaded through Matrigel and through the porous membrane at the bottom of the assay well. Scale bar 500 μm. **i** Quantification of the invasive potential of $p53^{fl/fl}$ tumour-derived organoids following control or Srsf1 shRNA manipulation, $n = 9$ vs 9 independent experiments. Data in bar charts are represented as mean and error bars are SD with data analysed with two-tailed, unpaired t-tests, p values are indicated in figure panels. All biological replicates are shown as individual value plots and $n > 3$. See also Fig. S6.

of gene recombination (depending on the experimental model), mice were induced with a single dose of tamoxifen (Sigma-Aldrich, T5648) by intraperitoneal injection of 120 mg/kg on the first day and then optionally with a further treatment of 80 mg/kg on the second and third days.

For the long-term Cre-Lox recombination in tumour cohorts, mice were given a 300 μL, 10 mg/mL dose of tamoxifen on day 0 and a 200 μL, 10 mg/mL dose of tamoxifen on day on day 1. Mice were then treated via intraperitoneal injection with azoxymethane (AOM) (Sigma-Aldrich, A5486) at a dose of 10 mg/kg once a week, every week, starting on day 12, for 6 weeks.

Mice were aged until symptomatic of disease (rectal bleeding, weight loss, hunching and/or pale feet). Mice were randomly distributed by sex and age and a minimum of 20 or 3 mice were used for long and short-term experiments respectively. In long-term tumour cohort mice, tumour number and burden were macroscopically quantified in situ after mouse termination.

For animals which were used for BrdU analysis, 200 μL of cell proliferation labelling reagent (GE Healthcare, RPN201) was administered via intraperitoneal injection 2 h before culling.

*Splicing factor-targeted synthetic lethal CRISPR screen & organoid transductions.* A targeted guide RNA (gRNA) library for the 60 APC-induced splicing factors was generated using sequences from Mouse GeCKOv2 Library A (Zhang Lab/GeCKO website), using 3 guide RNAs per gene and 9 non-targeting control guides (Supplementary Data 6). In order to avoid ligation bias, oligonucleotides (Sigma) corresponding to each guide RNA were cloned individually into the lentiGuide-Puro vector (Addgene, 52963), Sanger sequenced, and then pooled at equimolar concentrations. LentiGuide-Puro was a gift from Feng Zhang[60]. Large-scale plasmid prep was achieved by electroporation of Endura electrocompetent cells (Lucigen) as previously described (Zhang Lab/GeCKO website). Equal guide RNA coverage within the library was confirmed via next-generation sequencing (NGS). The guide RNA library was used to generate infectious lentiviral vector particles.

The workflow of the synthetic lethal screen was as follows: Day 0 – Wild type-Cas9 and $Apc^{fl/fl}$-Cas9 expressing organoids (derived from tamoxifen-inducible *villin*Cre^ERT2 LSL-Cas9 mice) were plated and grown (full 24-well plate each, 20 μl Matrigel/well) in organoid growth media supplemented with 6 μM CHIR-99021 (GSK-3 inhibitor, Abcam, ab120890). Day 3 – Organoids were further expanded to two full 24-well plates each and grown in 'Organoid+ media': ADF (Advanced DMEM/F12 + B27 + N2) (500 μL/well), Noggin, EGF, R-spondin (only for Wild type organoids), 10 μM Y-27632 dihydrochloride (ROCK inhibitor, Tocris, 1254), 6 μM CHIR-99021, 1 μM Jagged-1 (188–204) (Notch Ligand, AnaSpec, AS-61298) and 1 mM valproic acid (Sigma, PHR1061). Day 5 – organoids were dissociated via mechanical disaggregation followed by treatment with StemPro Accutase Cell Dissociation Reagent (Gibco, A1110501) for 5 min at 37 °C and then neutralised in 1% BSA. Cells were counted using the Countess II Automated Cell Counter (Invitrogen) and $1×10^6$ wild type-Cas9 or $Apc^{fl/fl}$-Cas9 organoid cells were transduced at a multiplicity of infection (MOI) of 0.3 for each of the three biological replicates. (The precise working titre of the virus was calculated in advance by viral titration experiments on non-Cas9 expressing $Apc^{fl/fl}$ organoids and measuring survival following puromycin selection). Briefly, 250,000 cells were plated on an 80 μL bed of Matrigel in each of four wells of a 12-well plate ($1×10^6$ cells per genotype). Cells were incubated with 0.3 MOI lentivirus and 4 μg/mL of Hexadimethrine bromide/polybrene (Sigma, H9268) in a total volume of 500 μL of organoid+ media. Cells were allowed to transduce for 24 h. Day 6 – virus was removed and adhered organoid cells were overlaid and set with 100 μL Matrigel, followed by standard organoid growth media with Y-27632 only.

Day 7 – Puromycin was added at a concentration of 2 μg/mL and Y-27632 was maintained in the media. Day 8 – fresh puromycin and Y-27632 containing media was added to organoids. Day 11 - puromycin and Y-27632 treatment was stopped and cells were grown in standard organoid growth media, and this was also refreshed on Day 13. Organoids were harvested on Day 14 and genomic DNA was purified using the DNeasy Blood & Tissue Kit (Qiagen, 69504). 4 independent replicate experiments were performed.

*Clonogenicity analysis.* Intestinal organoids were passaged as usual via mechanical disaggregation and DMEM/PBS washes. After the last wash step, organoid cell pellets were treated with 1 mL StemPro Accutase cell dissociation Reagent (Gibco, A1110501) and incubated at 37 °C for 5–10 min in order to generate single cells. An equal volume of 1% BSA was added to stop the digestion reaction and then diluted in 10 mL DMEM/F-12 media. Cells were then passed through a 40 μm cell strainer and centrifuged at 300 g for 3 min. Cells were counted using the Countess II Automated Cell Counter (Invitrogen). Between $1–10 × 10^3$ single cells were plated per 5 μL drop of Matrigel/BME and a minimum of 4 drops were plated for each genotype/condition. Organoid growth media was added and resultant spheres/clones were counted after 4 days. The clonogenic capacity was determined by calculating the average percentage of spheres formed in each drop per number of single cells plated.

*Dedifferentiation assay.* Mice were administered 120 mg/kg tamoxifen on day 0 and 80 mg/kg tamoxifen on days 1 and 2. On day 3, mice were culled and the first 10 cm of the small intestine following the duodenum was dissected and washed twice with PBS. The intestine was opened longitudinally with small scissors and the opened intestine was rinsed in PBS. Villi were removed by scraping using a glass coverslip, and were collected in DMEM/F12 in a 50 mL centrifuge tube. The tube was gently inverted 5–6 times in order to dissolve the mucus and debris was allowed to settle for 30 s. After the larger aggregates settled down, the supernatant containing the villi was decanted into another centrifuge tube. This supernatant was then centrifuged at 100 g for 3 min and the collected villi pellet was resuspended gently in 10 mL DMEM/F12 so as not to fragment or disrupt the villi structure. Whole villi were observed under a light microscope and then counted. Equal numbers of villi were taken for each genotype/condition, centrifuged at 100 g for 3 min and then resuspended in 3–5 mL TryplE Express (Gibco, 12605010) and incubated at 37 °C for 30 min. During the incubation, the suspension was vigorous resuspended via pipetting every 10 min. After each resuspension, a droplet of the digestion medium was observed under the microscope to check digestion. After digestion to single cells, cells were resuspended in 10 mL DMEM/F12 and passed through a 40 μm cell strainer. Single cells were counted and then 12 droplets of Matrigel/BME of 10 μL containing 50,000 cells per droplet were plated in organoid growth media, with the addition of 10 μM Y-27632 dihydrochloride (ROCK inhibitor, Tocris, 1254). Colonies/spheres resulting from dedifferentiated villi-derived single cells were scored after 7 days.

*RNAseq.* For the wild type vs $Apc^{fl/fl}$ and $Apc^{fl/fl}$ vs $Apc^{fl/fl}$ $Srsf1^{fl/+}$ RNAseq experiments, 3 mice from each genotype were given intraperitoneal injections of tamoxifen of 120 mg/kg on day 0 and 80 mg/kg tamoxifen on day 1. The mice in the $Apc^{fl/fl}$ vs $Apc^{fl/fl}$ $Srsf1^{fl/+}$ experiment received an additional 80 mg/kg tamoxifen induction on day 2. Mice were culled on day 5, and the small intestine was dissected and flushed with PBS. A 1 cm piece of small intestine following the duodenum was placed in RNAlater solution (Sigma). RNA was later extracted as previously described and was sent to Edinburgh Genomics for sequencing. Truseq (Illumina) mRNA-seq libraries were prepared from total RNA and these were then sequenced using the Illumina HiSeq 4000 using 150 base paired-end sequencing.

*Human organoid culture conditions.* Human adenoma, carcinoma and liver met organoids were cultured in: 30% R-spondin conditioned media, 1% Noggin conditioned media, 1x B27, 50 ng/mL EGF, 10 nM Gastrin, 10 nM PGE2, 10 mM Nicotinamide, 10 μM SB202190, 600 nM A83-01, 12.5 mM N-Acetylcysteine in ADF (with 1X PenStrep, HEPES, Glutamine). Human organoid transduction conditions were the same as for mouse organoid transductions except that single-cell suspension were generated in TrypLE with 10 μM Y27632 for 10 min at 37 °C with mechanical dissociation every 3 min.

*Human patient-derived organoids (PDOs).* C-002 is a cetuximab-resistant PDO generated from a biopsy of a liver metastasis from a gastrointestinal cancer

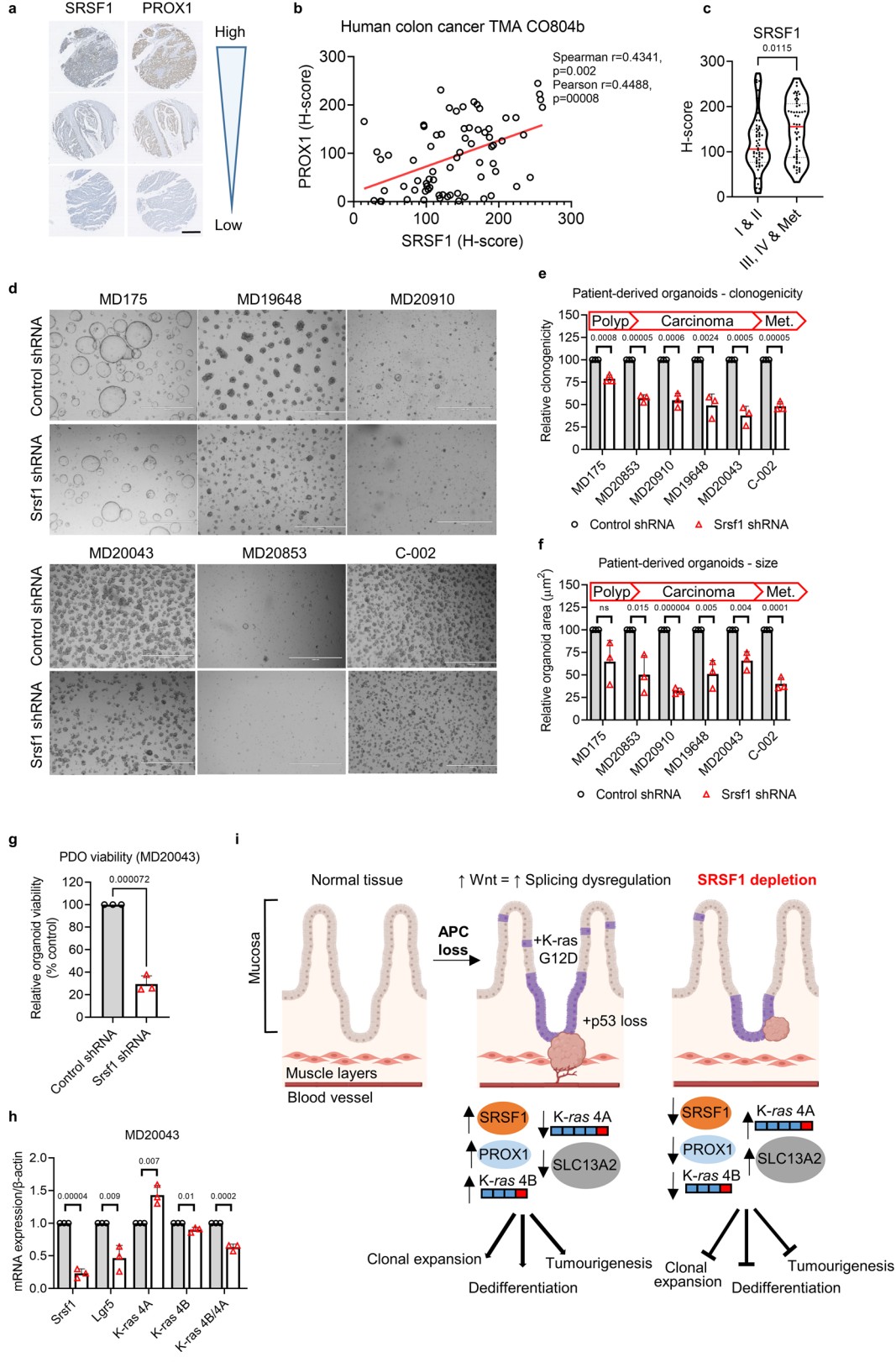

patient and was generated elsewhere[61]. The following PDOs were generated by Dr Farhat Din and Prof Malcolm Dunlop, University of Edinburgh: MD175 is a 50 yr old female with familial adenomatous polyposis who previously underwent a colectomy and ileorectal anastomosis who then developed a stage 2 rectal cancer - individual polyp used in this study. MD20043 is an 81 yr old male with stage 4 rectal cancer, TNM (T3, 2, 1). MD19648 is a 45-year-old female with familial adenomatous polyposis complicated by a stage 3 rectal cancer - FAP rectum tumour, TNM (pT1, pN1a, 0). MD20853 is a 71 yr old male with a tubulovillous adenoma with low-grade dysplasia. MD20910 is a 60 yr old male with rectal cancer, TNM (pT2, N1b, 0). Ethical approval for human CRC organoid derivation was carried out under NHS Lothian Ethical Approval Scottish Colorectal Cancer Genetic Susceptibility Study 3 (SOCCS3) (REC reference: 11/SS/0109). All patients provided fully informed consent for the use of their tissues.

**Fig. 7 Srsf1 depletion inhibits growth of human colorectal cancer-derived organoids. a** A sample of cores taken from a human colon cancer tissue microarray (TMA), with the same respective core stained for SRSF1 or PROX1 on different sections. Scale bar 500 μm. **b** Linear regression analysis showing the correlation of SRSF1 and PROX1 staining (based on histoscore) on the TMA shown in **a**, with each datapoint representing a core taken from a patient. **c** Relationship between SRSF1 immunohistochemistry staining and tumour stage in human patients, using TMA CO2081b, $n = 54$ vs 56 biologically independent tumour cores. **d** Representative images of patient-derived organoids (PDOs) treated with control or Srsf1 shRNAs. MD175 (polyp), MD20853 (CRC), MD20910 (CRC), MD19648 (FAP rectum Tumour), MD20043 (rectal carcinoma), C-002 (liver metastasis). Scale bar 1000 μm. **e** Number of surviving organoid clones after shRNA treatment and **f** size of indicated PDOs, $n = 3$ vs 3 independent experiments. **g** Viability of PDO MD20043 after Srsf1 shRNA treatment and **h** mRNA expression (qPCR) of indicated genes, $n = 3$ vs 3 independent experiments. **i** Model outlining the role of SRSF1 in modulating tumour cell plasticity and invasion in colorectal cancer. Data in bar charts are represented as mean and error bars are SD with data analysed with two-tailed, unpaired $t$-tests, $p$ values are indicated in figure panels. All biological replicates are shown as individual value plots and $n > 3$. See also Figure S7.

---

*Biological pathway enrichment analysis.* RNAseq data of the 2330 genes upregulated in $Apc^{fl/fl}$ relative to wild type were analysed using g:Profiler version e98_eg45_p14_ce5b097, accessed on 24/01/2020 using the following data sources: biological pathways-KEGG.

*Organoid Culture.* To generate organoids from wild type or genetically modified mice, the first 10 cm of small intestine following the duodenum was isolated, flushed with phosphate-buffered saline solution (PBS) and opened longitudinally using scissors. Villi were removed by scraping using a microscope coverslip. Remaining tissue was then washed several times with PBS and then incubated with 2 mM EDTA in PBS with gentle shaking at 4 °C for 30 min. Crypts were then removed from the tissue by vigorous pipetting and selecting through a 70 μm cell strainer. Crypts were scored and ~100 crypts were plated in 20 μL droplets of cold liquified Matrigel Growth Factor Reduced (GFR) Basement Membrane Matrix, Phenol Red-free, LDEV-free (Corning #356231/734-1101) or Cultrex Reduced Growth Factor Basement Membrane Extract Matrix, Type 2 (BME 2) (Trevigen #3533-010-02). A single 20 μl matrix droplet was grown in each well of a 24-well plate. Matrix droplets with suspended cells were allowed to solidify at 37 °C for 10 min prior to the addition of growth media. All organoids were grown in a humidified incubator at 37 °C supplemented with 5% $CO_2$.

Organoids were grown in 'organoid growth media': Advanced DMEM/F-12 media (ADF, Gibco) supplemented with 100 units/mL penicillin, 100 μg/mL streptomycin, 2 mM L-Glutamine and 10 mM HEPES (Life technologies), N2 (Gibco, 17502048), B27 (Gibco, 17504044), 50 ng/mL EGF (Peprotech, 315-09-500) and 100 ng/mL Noggin (Peprotech, 250-38-500). Wild type organoids were grow with the addition of 500 ng/mL R-spondin-1 (Peprotech, 120-38-500). We later used Noggin and R-spondin via way of conditioned media made by transformed HEK 293 cell lines. The transformed Noggin producing cell line was a gift from Hans Clevers' group (Hubrecht Institute) and the R-spondin producing cell line was purchased from Trevigen (#3710-001-01). Organoids were passaged via mechanical disaggregation using a serological pipette, where several rounds of ADF or PBS washing and centrifugation were used to remove the residual matrix, before resuspending organoid fragments in the fresh matrix.

*Tumour cell isolation and organoid culture.* Tumours were dissected from the intestine and then cut into pieces. Tumour tissue fragments were then incubated in 2 mL organoid growth media supplemented with 1 mg/mL collagenase II (Sigma, C1764), 0.5 mg/mL hyaluronidase (Sigma, H3506) and 10 μM Y-27632 dihydrochloride (ROCK inhibitor, Tocris, 1254). Incubation at 37 °C for 60–90 min with vigorous shaking until the tumour was completely disaggregated. The digested reaction was then neutralised using 100 μL 1% BSA and then fragments were filtered through a 70 μm cell strainer. The filtered cells were then centrifuged at 600 g for 3 min and washed 2–3 times in DMEM/F12, before being resuspended and plated in an 20 μl droplets of BME in 24-well plates and overlaid with organoid growth media.

*Pladienolide B treatment.* Pladienolide B stock (Calbiochem, 5.30196.0001) was made up to 1 mM in DMSO. Wild type and $Apc^{fl/fl}$ organoids per broken into fragments via mechanical disaggregation and were plated at a density of ~40 fragments/5 μL drop of Matrigel in a 96-well plate, with at least 6 wells per Pladienolide B concentration for each of the three biological replicates. Organoids were allowed 48 h to develop prior to the addition of Pladienolide B, and were treated for 48 h with concentrations of Pladienolide B ranging from 0.5 nM–200 nM. For human organoid experiments the same protocol was used but a single dose of 1 nM Pladienolide B was used.

*MTT/Resazurin cell viability analysis.* For Thiazolyl Blue Tetrazolium Blue (MTT) cell viability assay, organoids were plated and treated in 96-well plates as already described. MTT (Sigma, M2128) solution was added to the organoid culture media at a final concentration of 500 μg/mL and incubated for 2–3 h at 37 °C, 5% $CO_2$. The medium was then discarded and 20 μL of 2% SDS solution in H2O was added to solubilize the Matrigel (2 h, 37 °C). Then, 100 μL of DMSO were added for 1 h (37 °C) to solubilize the reduced MTT. The optical density was measured on a

microplate absorbance reader (Wallac, 1420 Victor2 microplate reader) at 570 nm and a background read at 690 nm was subtracted from this value. Cell viability was normalised to either Wild type or $Apc^{fl/fl}$ DMSO control vehicle-treated organoids.

For Resazurin cell viability assay, Resazurin (R&D systems, AR002) was added at a volume equal to 10% of the cell culture media volume and incubated for 4 h at 37 °C. Fluorescence was read using 544 nm excitation and 590 nm emission wavelength. For CRISPR gene-targeting assays, cell viability was assessed by first removing the fluorescence of puromycin treated, non-transduced (and therefore dead) organoids, followed by normalisation relative to either wild type or $Apc^{fl/fl}$ non-targeting guide RNA control transduced organoids.

*Overexpression of SRSF1 wild-type/DK mutant.* Srsf1WT and Srsf1DK were generated as above and cloned into the pUltra-GFP vector (Addgene, 24129). The pUltra was a gift from Malcolm Moore[62]. PCR amplificaton used 5'-XbaI and 3'-BamHI restriction sites, with the addition of a C-terminal FLAG tag. Transduction was performed as already described and cells were expanded and subsequently subjected to cell sorting (FACS) for GFP + cells. For the clonogenicity assay – cells were subjected to FACS and seeded at 5000 cells/10 μl BME drop in Y27632, and cells were counted 5 days following FACS sorting.

*Real-time quantitative polymerase chain reaction (qPCR).* Gene expression analysis was achieved via qPCR of mRNA. Total RNA was extracted using the RNeasy Mini Kit (Qiagen, 74106) followed by DNase treatment (Invitrogen, AM1906) and first-strand cDNA synthesis using 500 ng -1 μg RNA (Quantabio, 95048-100). qPCR was performed on 1/10 diluted cDNA using SYBR Select Master Mix (Applied Biosystems, 4472920) according to the manufacturer's instructions on the CFX Connect Real-Time System (Bio Rad). Gene-specific oligonucleotides used are listed in (Supplementary Data 14). Ct-values were normalized to β-Actin, GAPDH or 18srRNA Ct-values, or the geometric mean of a combination of these loading controls. The delta-delta Ct method was used in order to calculate the relative fold change in gene expression of samples.

*Reverse transcription polymerase chain reaction (RT-PCR) analysis of splicing isoforms.* RNA extraction and first-strand cDNA synthesis using 500 ng RNA were performed as previously described. A PCR reaction was then performed with gene-specific oligonucleotides (Supplementary Data 14) using Phusion High-Fidelity DNA Polymerase (NEB, M0530S) following the manufacturer's instructions using 5 μL of 1/10 diluted cDNA. PCR products were then separated by electrophoresis on a 1% agarose gel and fragment sizes were analysed using ImageJ[63] using GAPDH RT-PCR for loading normalisation. Splicing isoform changes were calculated by generating a ratio of expression of each respective isoform in each condition.

*CRISPR screen target validation experiments.* For validation of individual genes arising from the CRISPR screen, the most effective guide RNA (based on Z-score) was selected for gene targeting. Guide RNAs used for the validation were: Non-targeting#2 (5'-GCTTTCACGGAGGTTCGACG-3'), Srsf1 (5'-CGGGTCCTCGAA CTCAACGA-3'), Sf3b4 (5'-TTTAGATGCCACGGTGTACG-3') and Ddx10 (5'-AATATCACCTACTCGAGAAC-3'). Guide RNAs were individually cloned into the lentiGuide-Puro vector (Addgene) and transformed into Stbl3 chemically competent E. coli (Invitrogen, C737303). Wild type-Cas9 and $Apc^{fl/fl}$-Cas9 expressing organoids, as well as non-CAS9 expressing wild type and $Apc^{fl/fl}$ organoids, were transduced with lentivirus containing the specific guide RNA. This was done using 250,000 cells for each of the four cell genotypes. Two days after transduction, cells were selected with puromycin for 3 days. Surviving organoids were then digested to single cells using StemPro Accutase Cell Dissociation Reagent (Gibco, A1110501) and a clonogenicity assay was performed by plating 10,000 cells per 5 μl drop of Matrigel, with at least 4 Matrigel droplets per condition. Organoid growth media was added, with the addition of R-spondin-1 and 10 μM Y-27632 dihydrochloride (ROCK inhibitor, Tocris, 1254) to the Wild type-Cas9 and Wild type-Cas9-null organoids only. Four days after seeding, organoids resulting from single cells were scored and the clonogenic capacity was calculated as a percentage based on the number of cells seeded. Normalised clonogenicity values were then

calculated by first normalising the Cas9-expressing cells with the Cas9-null cells for each genotype, followed by normalising these values to the nontargeting guide RNA expressing condition.

For the cell viability assay in *Apc*<sup>fl/fl</sup> organoids, 10,000 single cells of either *Apc*<sup>fl/fl</sup>-Cas9 or *Apc*<sup>fl/fl</sup>-Cas9-null single cells were plated onto a bed of 20 μl Matrigel in a well in a 96-well plate in triplicate and were transduced with lentivirus-containing media. The day after transduction (day 1), lentivirus was removed and cells were overlayed with 20 μl Matrigel and media. On day 2 puromycin selection was started. On day 8, resazurin (R&D systems, AR002) was added to the media for 4 h at 37 °C and fluorescence was read using 544 nm excitation and 590 nm emission wavelength. Cell normalisation was done as previously described, first using the Cas9-null organoids, and then using the non-targeting guide RNA condition as a reference.

*Barcode PCRs and deep sequencing of integrated gRNAs.* Genomically integrated guide RNAs were PCR amplified using staggered, barcoded PCR primers as previously described by Zhang lab[64]. Sequencing of amplified PCR products was achieved by Illumina deep sequencing (MiSeq v2).

*Immunohistochemistry and histology.* Intestinal tissue was dissected and washed with PBS, and fixed either as a Swiss-roll or a parcel in formaldehyde 4% for 24 h at 4 °C. For tumour scoring, intestinal tissue was fixed in Methacarn (60% Methanol, 30% Chloroform and 10% Glacial acetic acid) for 5 min at room temperature. Tissue processing was done using the Tissue-TeK VIP infiltration Processor (Sakura) followed by embedding in paraffin. Tissue sections were prepared at 5 μm using a microtome (Leica). Standard immunohistochemistry & histology techniques were used. Harris Hematoxylin and Eosin Y were used for nuclei and cytoplasmic staining respectively (Thermo Fisher Scientific).

The following antibodies were used: BrdU (BD Biosciences, 347580, pH6, 1/500), PROX1 (R&D systems, AF2727, pH6, 1/100 for human tissue array, 1/200 for mouse sections), SLC13A2 (Atlas antibodies, HPA014963, pH8, 1/100), SRSF1 (Invitrogen/Thermo Fisher Scientific, 32-4600, pH8, 1/10,000), Ki67 (Abcam, ab15580, pH6, 1/2000), Active Caspase-3 (R&D systems, AF835, pH6, 1/800). Secondary detection was achieved using Dako EnVision + /HRP rabbit/mouse System, neat (Agilent technologies, K400311-2), except for PROX1 where Rabbit anti-Goat IgG was used 1/200 (Thermo Fisher Scientific, 81–1620). Staining was achieved by Diaminobenzidine (DAB) Quanto Chromogen and Substrate (Thermo Fisher Scientific).

To stain collagen, Picro Sirius Red was used: de-waxed slides were submerged in Picro Sirius Red solution for 2 h. Picro Sirius Red Solution: 0.1% Direct red 80 (Sigma, 41496LH), 0.1% Fast green FCF (Abcam, ab146267) in Picric acid solution 1.3% in H2O (Sigma, P6744). Post staining, slides were washed twice in acidified water, dehydrated according to standard protocols and mounted.

Images were taken by using either the BX53 Upright Microscope (Olypmus) with the CellSens imaging software (Olympus) or the Nanozoomer Digital slide scanner (Hamamatsu) with the NDP.view2 software (Hamamatsu). Image analysis and quantification was done using QuPATH[65].

The colon cancer human tissue arrays used were CO804b and CO2081b. (Biomax).

*shApc;KrasG12D/+;shRNA de-differentiation assay.* shApc;KrasG12D/+ organoids were previously described[36], and cultured in ADF + + media with 1X B27, 1X N2, EGF (50 ng/mL), 1% Noggin CM, 10 mM Nicotinamide, 1.25mM N-acetylcysteine. Organoids were lentivirally transduced with either pTripZ-minP-*shRenilla*, or pTripZ-minP-*shSrsf1*. Organoids were given either a 12 hr or 24 hr treatment with or without doxycycline (1 μg/mL) before being collected and digested to single cells in TrypLE containing 10 μM Y-27632 for 30 min at 37 C. Cells were washed in ADF + + before being passed through a 40 μM cell strainer. Single cells were centrifuged at 500x *g* for 5 min at 4 C before being stained with 1:200 EphB2-APC conjugated antibody (BD Bioscience, Clone 2H9) for 30 min at room temperature in PBS with 0.1% BSA and 10 μM Y-27632. Cells were then washed twice in PBS at 500x *g* for 5 min at 4 C before being subjected to FACS (BD FACSARIA™ III) for EphB2low and EphB2high (no doxycycline) or GFP + /RFP + /EphB2<sup>low</sup> and EphB2<sup>high</sup> (doxycycline) populations. Single cells were then seeded into 20 μl BME in organoid media containing 10 μM Y-27632 for 48 h. Formed organoids were counted 7 days after FACS, and clonogenicity was calculated as (#GFP + /RFP + organoids/#single cells seeded x100).

*EPHB2 negative in vivo dedifferentiation assay.* Mice were administered 120 mg/kg tamoxifen on day 0. On day 2, mice were culled and the first 10 cm of the small intestine following the duodenum was dissected and washed twice with PBS. The intestine was opened longitudinally with small scissors and the opened intestine was rinsed in PBS. Villi were removed by scraping using a glass coverslip, and were collected in DMEM/F12 in a 50 mL centrifuge tube. The tube was gently inverted 5–6 times in order to dissolve the mucus and debris was allowed to settle for 30 s. After the larger aggregates settled down, the supernatant containing the villi was decanted into another centrifuge tube. This supernatant was then centrifuged at 100 g for 3 min and the collected villi pellet was resuspended gently in 10 mL DMEM/F12 so as not to fragment or disrupt the villi structure. Whole villi were observed under a light microscope and then counted. Equal numbers of villi were taken for each genotype/condition, centrifuged at 100 g for 3 min and then

resuspended in 2 mL TryplE Express (Gibco, 12605010) and incubated at 37°C for 30 min. During the incubation, the suspension was vigorous resuspended via pipetting every 10 min. After each resuspension, a droplet of the digestion medium was observed under the microscope to check digestion. After digestion to single cells, cells were resuspended in 10 mL DMEM/F12 and passed through a 40 μm cell strainer. Single cells were washed once with 0.1% BSA in PBS and stained with 1:200 EphB2-APC conjugated antibody (BD Bioscience, Clone 2H9) and 1:200 Epcam-PE-conjugated antibody (BD Biosciences, 563477) for 30 min at room temperature in PBS with 0.1% BSA. Cells were then washed twice in 0.1% BSA in PBS at 500 g for 5 min at 4 C before being subjected to FACS (BD FACSARIA™ III) for Epcam positive, EphB2 negative populations. Single cells were then seeded into 10 μl BME in organoid media containing 10 μM Y-27632. Formed organoids were counted 7 days after FACS, and clonogenicity was calculated as (#organoids/#single cells seeded x100).

*LGR5-GFP clonogenicity.* vil-Cre-ERT2 WT and *Srsf1*<sup>fl/+</sup> mice carrying the *Lgr5GFP-CRE*<sup>ERT2</sup>transgene were administered 120 mg/kg tamoxifen on day 0 and 80 mg/kg tamoxifen on day 1. On day 7 mice were culled and the first 10 cm of the small intestine following the duodenum was dissected and washed twice with PBS. The intestine was opened longitudinally with small scissors and the opened intestine was rinsed in PBS. Villi were removed by scraping using a microscope coverslip. The remaining tissue was then washed several times with PBS and then incubated with 2 mM EDTA in PBS with gentle shaking at 4 °C for 30 min. Crypts were then removed from the tissue by vigorous pipetting and selecting through a 70 μm cell strainer. Equal numbers of crypts were taken for each genotype/condition, centrifuged at 100 g for 3 min and then resuspended in 2 mL TryplE Express (Gibco, 12605010) and incubated at 37 °C for 15 min. During the incubation, the suspension was vigorous resuspended via pipetting every 5 min. After each resuspension, a droplet of the digestion medium was observed under the microscope to check digestion. After digestion to single cells, cells were resuspended in 10 mL DMEM/F12 and passed through a 40 μm cell strainer. Cells were washed twice in 0.1% BSA in PBS at 500 g for 5 min at 4 C before being subjected to FACS (BD FACSARIA™ III) for the Lgr5-GFP positive population. Single cells were then seeded into 10 μl BME in organoid growth media containing 10 μM Y-27632. Formed organoids were counted 7 days after FACS, and clonogenicity was calculated as (#organoids/#single cells seeded x100).

*Tumour stage classification.* Staging of tumours was scored according to the TNM staging system; T0- no evidence of primary tumour, Tis- carcinoma in situ, T1-tumour invading submucosa, T2- tumour invading muscularis propria, T3-tumour invading through the muscularis propria into the pericolorectal tissue, T4a- tumour penetrating to the surface of the visceral peritoneum, T4b- tumour directly invading or adherent to other organs or structures.

*BrdU cell counting.* Images of stained small intestinal parcels were analysed in ImageJ where villi and crypts were first divided into villi and crypt compartments. BrdU positive cells in each compartment were scored from several images analysing an average of 24 villi/mouse and 82 corresponding crypts/mouse, using 3–5 mice per genotype.

*cDNA constructs and SRSF1 mutagenesis.* Srsf1 cDNA was purchased from GeneCopoeia (EX-Mm13449-Lv122) and Prox1 cDNA was from Origene (MR210370). Kras 4 A and Kras 4B were cloned from cDNA from *Apc*<sup>fl/fl</sup> mice.

The *Srsf1* RNA recognition motif 2 (RRM2) mutant 'DK' was created by forcing amino acids 136D and 138 K both to mutate to alanine. This was achieved by conducting a mutagenesis PCR on *Srsf1* within the Lv122 vector using the following overlapping oligonucleotides: 5'-CAAGTGGAAGTTGGCAGGCTTTAG CGGATCACATGCGTGAAG-3' and 5'-CTTCACGCATGTGATCCGCTAAAGC CTGCCAACTTCCACTTG-3'.

*SRSF1 RNA immunoprecipitation Protocol.* 10 million CMT93 cells were pelleted and washed twice with PBS. Cells were homogenised in 1 ml lysis buffer, pH 7.4 (100 mM NaCl, 10 mM MgCl2, 30 mM Tris-HCl, 1 mM DTT, Protease & phosphatase inhibitor, 40 U/ml RNase OUT, 0.5% Triton X-100). Homogenate was centrifuged at 10,000 g for 10 min and 100ul of supernatant reserved for RNA and protein input samples. The remaining lysate was split in 2 with half used for IgG IP and half for SRSF1 IP. The lysate was precleared with 25ul washed Dynabeads, 0.05% BSA and 0.1ug/ml yeast tRNA for 1 h, 4 degC. Lysate was moved to a fresh tube and incubated with 2 ug SRSF1 antibody (sc-73026) or 2 ug mouse IgG (ab37335) for 1 h 4 degC pre-incubation with rotation, followed by a further 2 h incubation of 50ul dynabeads. The lysate was discarded and beads washed 3 x with lysis buffer. 1/3rd of the beads were taken for Western blots and the remaining resuspended in 100ul lysis buffer, followed by the addition of 50ug Proteinase K. Beads were incubated for 15 min 37 degC to elute RNA from beads. RNA was purified in Trizol and purified RNA reverse transcribed. cDNA was used for qRT-PCR reactions to determine relative binding of SRSF1 to various RNA transcripts.

*BioID protein interaction screen.* The pUltra vector (Addgene, 24129)[62] was first modified, swapping out the EGFP for a gene encoding puromycin resistance. DNA

sequence encoding the BirA enzyme with an N-terminal Myc tag (Myc-BioID2-MC, Addgene, 74223) was then subcloned into the pUltra vector (Addgene, 24129). Myc-BioID2-MCS was a gift from Kyle Roux[66,67]. Finally, primers containing the sequences of EcoRV and EcoRI restriction enzymes were used for Kras 4 A and Kras 4B amplification. Primer sequences were Kras 4 A/4B forward: 5′- CTCTCTG ATATCGACTGAGTATAAACTTGTGGTGGTTGGAGCTGGTGGCGTAG -3′, Kras 4 A reverse: 5′-AGAGAGGAATTCTTACATTATAAACGCATTTTTTAATT TTCACACAGCCAGGA-3′and Kras 4B reverse: 5′-AGAGAGGAATTCTCACA-TAACTGTACACCTTGTCCTTGACTTCTTCTTCTTC-3′. For the purpose of a BioID interaction control protein, the gene encoding GFP was cloned into the BirA-pUltra vector.

The BioID streptavidin affinity purification experiment was based on a previously published protocol[66]. In the first instance, 10 cm² dishes were seeded each with $2 \times 10^6$ CMT-93 cells (mouse rectal carcinoma), 24 h prior to transfection. CMT-93 cells were grown in Dulbecco's Modified Eagle's Medium (DMEM, Sigma) supplemented with 10% foetal bovine serum, 2 mM glutamine and 1% of penicillin-streptomycin. Four plates were used for each of the three conditions: BirA-GFP, BirA-Kras4A and BirA-Kras4B. The following day, 8 μg of plasmid DNA was used for transfection in each plate using Lipofectamine 2000 (Invitrogen, 11668030) and Opti-MEM (Gibco, 51985026) according to the manufacturer's instructions. 24 h later, growth media was replaced and supplemented with 50 μM biotin (Sigma, B4501). Cells were incubated for 24 h, washed in PBS, and then scraped into 500 μL RIPA buffer. Following protein clarification, 25 μL Streptavidin Sepharose slurry (GE Healthcare, 17-5113-01) was washed in RIPA buffer before being added to 1 mL of protein lysate, which was incubated for 6 h at 4 °C with rotation. Beads were purified by centrifugation at 1000 g for 5 min and washed four times with buffer (50 mM TrisCl and 8 M Urea, pH 7.4). On-bead digestion was done with trypsin as previously described[68], followed by mass spectrometry using the Fusion Lumos mass spectrometer (Thermo Fisher Scientific). Proteins analysis was done using the MaxQuant-Perseus software[69], and proteins were mapped to the mouse Uniprot database. Proteins enriched in BirA-Kras4A and BirA-Kras4B vs BirA-GFP were determined by first generating a Log2 of the LFQ values and then normalising by Z-score. A student's t-test (p < 0.05) was performed versus the GFP control samples. Significant hits were also filtered by two additional parameters: >1.5 fold higher levels in Kras 4A/4B expressing cells and at least 2 peptides detected in each of the three biological replicates. Cluster analysis was done using STRING[70].

Western blotting. Cells were lysed using RIPA buffer (Sigma, R0278) supplemented with 1% of phosphatase and protease inhibitors (Sigma, P0044 and P8340). Protein concentration was calculated using the BCA Protein Assay kit (Pierce). 10 μg of denatured protein lysate was separated by electrophoresis on NuPAGE 4–12% Bis-Tris precast protein polyacrylamide gels (Invitrogen) and blotted onto 4 μm nitrocellulose membrane (Amersham). Membranes were immersed in blocking solution (5% milk, 0.1% PBS-tween) for 1 h at room temperature, before being incubated in primary antibody at 4 °C overnight. Following 3 washes in 0.1% PBS-tween, the membrane was incubated in secondary antibody for 1 h at room temperature, followed by 3 more PBS-tween washes. Antibody signal was detected by using ECL Plus Western blotting substrate (Pierce, 32132) and visualised using the ImageQuant LAS 4000 (GE Healthcare). Primary antibodies used were: β-actin, 1/5000 (Cell Signalling Technology, 4970), SF2/SRSF1, 1/1000 (Abcam, ab133689), c-Myc, 1/1,000 (Cell Signalling Technology, 9402), Prox1, 1 μg/mL (R&D systems, AF2727), Ras (27H5), 1/1000 (Cell Signalling Technology, 3339), Myc-Tag (9B11), 1/10,000 (Cell Signalling Technology, 2276), and Streptavidin HRP, 1/10,000 (Abcam, ab7403). Secondary antibodies used were: Anti-Rabbit IgG HRP-linked, 1/1000 (Cell Signalling Technology, 7074), Anti-Mouse IgG HRP-linked, 1/1000 (Cell Signalling Technology, 7076) and Rabbit anti-Goat IgG, 1/1000 (Thermo Fisher Scientific, 81–1620). Western blot signal densitometry analysis was done using ImageJ[63].

Lentiviral particles. Lentiviral particles were made using cloned plasmid vectors, either at the Shared University Research Facility (University of Edinburgh), or in our own laboratory. Briefly, 10 μg gene-specific lentiviral vector was mixed with 7.5 μg lentiviral packaging vector psPAX2 (Addgene) and 2.5 μg envelope protein producing vector pCMV-VSV-G (Addgene) and transfected into HEK 293 T cells in a 10 cm² dish using polyethylenimine transfection reagent (Polysciences, 23966). After 48 h virus was purified by first filtering the supernatant media with a 0.45 μm filter, followed by virus concentration using Lenti-X Concentrator (Takara Bio, 631232) and resuspension of viral particles in PBS.

Diagrams & illustrations. Diagrams were created with Biorender.com (Figs. 4G, 6G, 7I, S3R, S5H, S5Q, S5R, S6A and S6K) or in Microsoft PowerPoint.

Deltarasin treatment. Organoids were mechanically dissociated by vigorous pipetting. 100 fragments were seeded in 10 μl BME in a 24-well plate. 24 h after seeding, media was removed and replaced with fresh media containing either Deltarasin (Tocris, 5424) (10 μM) or DMSO. Cell proliferation was assessed after 48 h adding 10% Resazurin (R&D systems, AR002). Fluorescence was measured using Victor2 Multilabel Plate Reader (PerkinElmer) as already described.

Kras 4b morpholino treatment. Organoids were mechanically dissociated by vigorous pipetting. 20–100 fragments were seeded in 10 μl BME in a 24-well plate. 24 h after seeding, media was removed and replaced with fresh media containing either 5 μM Kras4B vivo-morpholino (Genetools - GTATAGAAGGCATCGTCA ACACCCT) or 5 μM control vivo-morpholino (Genetools - CCTCTTACCTCAG TTACAATTTATA). Cell proliferation was assessed after 6 days later by adding 10% Resazurin (R&D systems, AR002). Fluorescence was measured using Victor2 Multilabel Plate Reader (PerkinElmer) as already described.

Organoid invasion assay. The invasion assay was based on a previously described protocol[71]. FluoroBlok HTS 24 Multiwell Insert System with 8.0 μm Pore High Density PET Membrane (Scientific Laboratory Supplies Ltd., 351157) was used. Inserts were coated with 50 μg/mL Matrigel solution diluted in cooled DMEM/F12 media containing EGF and Noggin. The organoids were dissociated with TryplE (Gibco, 12605010), resuspended in DMEM/F12 media containing EGF, Noggin and 5 ng/ml TGF beta-1 (PeproTech EC Ltd, 100-21-10) and $3 \times 10^4$ cells were seeded in each apical chamber of the Fluoroblock insert, while DMEM/F12 media containing EGF and Noggin was added in the basal chambers. Following 72 h incubation at 37 °C, 5% CO₂ atmosphere, cells were stained for 1 h with Calcein AM (Abcam, ab141420) and bottom and top fluorescence was read on a Victor2 Multilabel Plate Reader (PerkinElmer) at wavelengths of 485/535 nm (Ex/Em).

Bioinformatics. Differential Gene Expression of Wild type vs $Apc^{fl/fl}$ and $Apc^{fl/fl}$ vs $Apc$;$^{fl/fl}$ $Srsf1$:$^{fl/+}$ Adaptor sequences were removed from paired reads using TrimGalore and cutadapt an8d reads mapped to mouse genome GRCm38 (mm10) using tophat 2.1.1[72]. Differential expression was called using cuffdiff 2.2.1[73] making use of Ensembl gene annotation 84 for genes (for Wild type vs $Apc^{fl/fl}$) and Ensembl gene annotation 87 for genes (for $Apc^{fl/fl}$ vs $Apc$;$^{fl/fl}$ $Srsf1$$^{fl/+}$). Genes with an adjusted p value of less than 0.05 were considered significantly regulated.

Splicing analysis Wild type vs $Apc^{fl/fl}$ and $Apc^{fl/fl}$ vs $Apc$;$^{fl/fl}$ $Srsf1$$^{fl/+}$ Paired reads were pseudoaligned to the GRCm38 (mm10) Ensembl 87 transcriptome using salmon v0.9.1[74]. Splicing changes were inferred from transcript TPMs using SUPPA2[20] from gene definitions in the Ensembl 87. A second splicing assessment was made with rMATS[21] using the —novelSS setting to obtain splicing events involving unannotated splice sites.

Gene Set Enrichment analysis (GSEA): Enrichment analysis of differentially expressed genes in the $Apc^{fl/fl}$ vs $Apc$;$^{fl/fl}$ $Srsf1$$^{fl/+}$comparison in gene sets defining enterocyte, paneth, goblet, tuft and enteroendocrine cell types was performed using the GSEA and enricher functions of Clusterprofiler[75]. The gene sets for these cell types was compiled previously[29]. Enrichment of differentially expressed genes in ISC, proliferating, late TA and Lgr5 gene sets, defined previously[30], was performed similarly after mapping genes across species using homologs specified in biomart.

Splicing factor targeted CRISPR screen wild type vs $Apc$:$^{fl/fl}$ A library of 180 gRNAs targeting 60 genes (plus 9 non-targeting controls) was designed using Mouse GeCKOv2 Library A (Supplementary Data 6). Stagger sequences were removed from reads using cutadapt prior to read counting and statistical testing of counts in treatment vs control using MAGeCK v0.5.6[76]. Statistics for gRNAs were used to select genes with gRNA having increased counts in treatment.

Enrichment of splicing events: An enrichment algorithm for the comparison of an ordered set (A) of splicing events with another (B) was implemented following the gene set enrichment method[77]. Splicing events in the reference set A were ordered by p value (for events with p < 0.1) and absolute dPSI used to associate a probability with each event. Events common to a test set B and A were considered hits and increased a cumulative enrichment score by the probability of each matching event, non-hit events decreased the score. The observed enrichment was the maximum of this score. The maximum enrichment possible was where the number of intersecting events occurred at the head of A, and a probability value for the actual enrichment was calculated from a null distribution of maximum scores found by randomly selecting 100,000 subsets of A with the size of the intersection of A and B.

Mouse shRNA constitutive & Tet-On system and organoid size experiments. For constitutive knockdown, the control (scramble) shRNA was a gift from David Sabatini (Addgene, 1864)[78].

TRC lentiviral shRNAs (Mouse Srsf1 ENTREZGENE:110809) were purchased from Dharmacon. These were supplied in the pLKO.1 vector: shSrsf1-3 Clone Id: TRCN0000109142, sequence: AACATCAGCGTAACATACATC.

For inducible knockdown, shRenilla and shSrsf1−4 were cloned into Tet-ON inducible vector pTRIPZ, modified with a synthetic minimal promoter (minP) (synthesised by GenScript) to replace the minimum CMV promoter. shRNAs were cloned into pTripZ vector using 5′ XhoI and 3′EcoRI restriction sites. Organoids were seeded into BME and treated with 1 μg/ml doxycycline (Alfa Aesar by Thermo Scientific – J60579.22) dissolved in water.

shSrsf1−4 sequence (97mer): TGCTGTTGACAGTGAGCGATAGGCTTATGTTTGAACACTATAGTGAAGC-CACAGATGTATAGTGTTCAAACATAAGCCTACTGCCTACTGCCTCGGA.

shRenilla sequence: (97mer): TGCTGTTGACAGTGAGCGCGAGGAATTATAATGCTTATCTATAGTGAAGC-CACAGATGTATAGATAAGCATTATAATTCCTATGCCTACTGCCTCGGA.

Images were taken at the indicated time points and analysed using ImageJ 2.0. For organoid size (area) determination for the constitutive shRNA model, the minimum number of organoids used to calculate organoid area per biological replicate were: $Apc^{fl/fl}$ 15, AKP: 20 and KPN: 40. For tracking and size determination in the Tet-On models, the number of organoids measured & tracked per biological repeat were: Wild-type organoids: 5–17, $Apc^{fl/fl}$ organoids: 8–20 and AKP organoids: 5. For the shRNA $Srsf1/Kras$ 4B rescue experiments in $Apc^{fl/fl}$ organoids, 14–22 organoids were measured for size per biological replicate. For the human PDO organoids, the number of organoids measured were 33–250. All organoid size experiments had $n = 3$ biological replicates.

*Human shRNAs*. TRC lentiviral shRNAs (Human SRSF1 ENTREZGENE: 6426.) were purchased from Dharmacon. These were supplied in pLKO.1 vectors. shSRSF1–2 Clone ID: TRC0000001094, Sequence: TTAACCCGGATGTAG GCAGT.

*Quantification and statistical analysis*. Statistical analyses were performed using GraphPad Prism software (v8.3 GraphPad software, La Jolla, CA, USA) and Microsoft Excel (v2016, Redmond, WA, USA) performing the tests as indicated in the figure legends or main text. Significance levels were calculated according: $p < 0.05$ (*), $p < 0.01$(**) and $p < 0.001$ (***).

**Reporting summary**. Further information on research design is available in the Nature Research Reporting Summary linked to this article.

## Data availability

The RNAseq data generated in this study have been deposited in the GEO database under accession code GSE199623 (https://www.ncbi.nlm.nih.gov/geo/query/acc.cgi?acc= GSE199623) and are freely available. All data are provided within the article, Supplementary Information and source data.

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

## Acknowledgements

This work was funded by Cancer Research UK (CRUK) under Career Development Fellowship, A19166 (K.B.M) and Small Molecule Drug Discovery Project Award, A25808 (K.B.M) and the European Research Council under Starting Grant, COLGENES – 715782 (K.B.M). Mass spectrometry was supported by the Wellcome Trust (Multiuser Equipment Grant 208402/Z/17/Z). We thank the University of Edinburgh's Institute of Genetics and Cancer (IGC) technical staff for providing support for some of the experiments and we thank the animal technicians at the Biomedical Research Facility (BRF) facility for animal husbandry support. We thank Prof Owen Sansom for providing us with the VilCreERT2, Apc$^{fl}$, P53$^{fl}$, Kras$^{G12D}$ and Lgr5-CreERT2 mouse lines. We thank Dr Lukas Dow and Prof Scott Lowe for providing the shApc organoid line. We thank Prof Nicola Valeri for providing the C-002 PDO line.

## Author contributions

The project was conceived and experiments planned by A.E.H. and K.B.M. Experiments were conducted by A.E.H., S.O.P., M.R., P.C., A.B., P.P, N.T.Y and K.B.M. Bioinformatics analysis was conducted by S.A. C.V.B. helped with tissue histology and lentiviral preparation. A.B.C. and A.V.K conducted the mass spectrometry and analysis. P.F., M.D. and F.D. generated and provided the patient-derived organoids. F.H., I.R.A and J.F.C generated the Srsf1-NRS mouse. All aspects of the study were supervised by K.B.M. The manuscript was prepared by A.E.H. and K.B.M. and all authors read and approved it.

## Competing interests

The authors declare no competing interests.
