## [Peer Review File · Nature Communications]

RNA splicing is a key mediator of tumour cell plasticity and a therapeutic vulnerability in colorectal cancerREVIEWERS' COMMENTS

Reviewer #1 (Remarks to the Author):

The authors have addressed my criticisms. This revised manuscript has improved substantially and in my opinion, should be considered for publication in Nature Communications.

Reviewer #2 (Remarks to the Author):

The authors have responded to my initial comments and questions well and I have no further issues with the manuscript.

Reviewer #3 (Remarks to the Author):

In their revised version, the authors have convincingly addressed my concerns. New data is shown that supports the tumor-specific effects after Srsf1 reduction and Deltarasin/Pladienolide B treatment, which had also been remarked by reviewer 1. Furthermore, the motivation for analyzing heterozygous Srsf1 mutants is now explained clearer. The CRISPR screening data has been properly replicated as requested and the statistical analysis and reporting of this data has substantially improved. Together, I have no further comments. I thank the authors for their careful revision and congratulate them on this important work.

Reviewer #1 (Remarks to the Author):

The authors have addressed my criticisms. This revised manuscript has improved substantially and in my opinion, should be considered for publication in Nature Communications.

Reviewer #2 (Remarks to the Author):

The authors have responded to my initial comments and questions well and I have no further issues with the manuscript.

Reviewer #3 (Remarks to the Author):

In their revised version, the authors have convincingly addressed my concerns. New data is shown that supports the tumor-specific effects after Srsf1 reduction and Deltarasin/Pladienolide B treatment, which had also been remarked by reviewer 1. Furthermore, the motivation for analyzing heterozygous Srsf1 mutants is now explained clearer. The CRISPR screening data has been properly replicated as requested and the statistical analysis and reporting of this data has substantially improved. Together, I have no further comments. I thank the authors for their careful revision and congratulate them on this important work.

We thank the reviewers for taking the time to review our manuscript and their supportive comments to recommend publication.